# Wide-range continuous tuning of the thermal conductivity of La$_{0.5}$Sr$_{0.5}$CoO$_{3-\delta}$ films via room-temperature ion-gel gating

Yingying Zhang [1], William M. Postiglione [2], Rui Xie [3], Chi Zhang [1], Hao Zhou[3], Vipul Chaturvedi [2], Kei Heltemes [2], Hua Zhou [4], Tianli Feng [3], Chris Leighton [2] ✉ & Xiaojia Wang [1] ✉

Solid-state control of the thermal conductivity of materials is of exceptional interest for novel devices such as thermal diodes and switches. Here, we demonstrate the ability to *continuously* tune the thermal conductivity of nanoscale films of La$_{0.5}$Sr$_{0.5}$CoO$_{3-\delta}$ (LSCO) by a factor of over 5, via a room-temperature electrolyte-gate-induced non-volatile topotactic phase transformation from perovskite (with $\delta \approx 0.1$) to an oxygen-vacancy-ordered brown-millerite phase (with $\delta = 0.5$), accompanied by a metal-insulator transition. Combining time-domain thermoreflectance and electronic transport measurements, model analyses based on molecular dynamics and Boltzmann transport equation, and structural characterization by X-ray diffraction, we uncover and deconvolve the effects of these transitions on heat carriers, including electrons and lattice vibrations. The wide-range continuous tunability of LSCO thermal conductivity enabled by low-voltage (below 4 V) room-temperature electrolyte gating opens the door to non-volatile dynamic control of thermal transport in perovskite-based functional materials, for thermal regulation and management in device applications.

Perovskite oxides with nominal general formula ABO$_3$ are well known for their immensely tunable structures and compositions, and thus physical and chemical properties, making them attractive for applications in superconductivity[1], catalysis[2], various memory devices[3], solid oxide fuel cells[4], etc. A and B site cations can be readily substituted to tailor the crystal structure, electronic structure, and doping, while oxygen stoichiometry (e.g., oxygen deficiency $\delta$) adds additional control, particularly via oxygen vacancies (V$_O$s). Recently, *active* control of the oxygen stoichiometry in perovskites has come into the spotlight, enabling reversible voltage control of electronic, magnetic, and optical properties[5–9], with much technological potential. Electrochemical operating mechanisms[5], often present in electrolyte-gated perovskite-based devices employing ionic liquid or gel electrolytes, provide one powerful approach to this, where perovskite cobaltites have emerged

as prototypical targets. In strontium cobaltite, SrCoO$_{3-\delta}$, for example, a reversible non-volatile topotactic phase transformation between perovskite (P) SrCoO$_{3-\delta}$ (with randomly distributed V$_O$) and oxygen-vacancy-ordered brownmillerite (BM) SrCoO$_{2.5}$ has been achieved through electrolyte gating[10–12]. Due to the highly contrasting electronic and magnetic ground states of these two phases, voltage-driven cycling drives metal-insulator and ferromagnetic-antiferromagnetic transitions, as well as large changes in optical properties, drawing increasing attention[10–13]. Recent work by some of us established that such phenomena are in fact possible throughout the entire La$_{1-x}$Sr$_x$CoO$_{3-\delta}$ phase diagram, providing control over the threshold voltage for the P to BM transformation (via $x$), and circumventing issues with the air stability of P-SrCoO$_3$ (La$_{0.5}$Sr$_{0.5}$CoO$_3$, for example, is much more air-stable)[14]. In general, the electronic and magnetic properties of La$_{1-x}$Sr$_x$CoO$_{3-\delta}$ have

---

[1]Department of Mechanical Engineering, University of Minnesota, Minneapolis, MN, 55455, USA. [2]Department of Chemical Engineering and Materials Science, University of Minnesota, Minneapolis, MN, 55455, USA. [3]Department of Mechanical Engineering, University of Utah, Salt Lake City, UT, 84112, USA. [4]Advanced Photon Source, Argonne National Laboratory, Lemont, IL, 60439, USA. ✉e-mail: leighton@umn.edu; wang4940@umn.edu

been extensively explored and manipulated via electrolyte gating, via both electrostatic and electrochemical mechanisms[6,14–16].

While the above focuses on structural, electronic, magnetic, and optical properties, electrolyte gating is also attractive for voltage-based control of thermal properties, and associated applications. The latter include thermal diodes, thermal switches, and energy conversion and storage[17,18], wherein the thermal conductivity of materials is dynamically tuned to achieve desired performance. For example, Cho et al. demonstrated reversible voltage-based tuning of the thermal conductivity of the battery cathode material $Li_xCoO_2$, from 3.7 to 5.4 W m$^{-1}$ K$^{-1}$, via electrochemical modulation[19]. More recently, Lu et al. voltage tuned the thermal conductivity of $SrCoO_{3-\delta}$ by a factor of ~2.5 by transforming (oxidizing) as-deposited BM-$SrCoO_{2.5}$ to P-$SrCoO_{3-\delta}$, and by a further factor of ~4 by transforming (reducing) BM-$SrCoO_{2.5}$ to a hydrogenated H-$SrCoO_{2.5}$ phase, both via non-volatile electrolyte gating[10]. Notably, the factor of 4 tuning ratio of thermal conductivity via a bi-state (BM phase → H-containing phase) single-step tuning process appears to be a maximum among literature experimental studies[10,19–28]. While groundbreaking, it should be noted, however, that this work was not able to achieve continuous control of $\delta$ between P-$SrCoO_{3-\delta}$ and BM-$SrCoO_{2.5}$ by room-temperature (RT) electrolyte gating, that reversibility and speed remain open questions (particularly when H is involved), and that electronic contributions to the thermal conductivity tuning were found to be negligible, limiting the magnitude of the thermal conductivity tuning ratio[10].

In order to explore the limits of voltage-based tuning of thermal conductivity via electrolyte gating between P and BM phases, in this work we focus on $La_{0.5}Sr_{0.5}CoO_{3-\delta}$ (LSCO) as a model system. Critically, this system enables electrolyte gating from an as-deposited P phase with pristine metallic electrical conductivity, to an insulating BM phase, maximally modulating the electrical resistivity, with excellent air stability in both phases[14]. Such an approach has *not* been reported with $SrCoO_{3-\delta}$. We combine time-domain thermoreflectance (TDTR) measurements with high-resolution X-ray diffraction (XRD) and electronic transport measurements to explore the thermal conductivity tuning of LSCO through the topotactic P → BM transformation as a function of gate voltage ($V_g$). The $V_g$-driven cubic P to orthorhombic BM transformation and associated metal-insulator transition are found

to enable continuous, RT tuning of the thermal conductivity of LSCO films by a remarkable factor of over 5, a record value for such an approach. Combining temperature (T)-dependent thermal measurements with model analyses based on molecular dynamics (MD) simulations and the Boltzmann transport equation (BTE), and comparing to accompanying electrical transport measurements, we further establish the scattering mechanisms of heat carriers that enable this tuning of thermal conductivity. The relative impacts of the Sr substitution, the $V_g$-driven metal-insulator transition (i.e., the electronic contribution to the thermal conductivity), $\delta$ variation, crystal structure modification, and lattice symmetry change, are all discussed in detail.

## Results
### Structural characterization of the P → BM phase transformation
In this work, ~45-nm-thick films of P-LSCO epitaxially deposited on $LaAlO_3(001)$ (LAO) substrates were driven to the $V_O$-ordered BM structure (BM-LSCO, $\delta \sim 0.5$) by removing oxide ions ($O^{2-}$) using an all-solid-state side-gated electrochemically operating electrolyte-gate device, shown schematically in Fig. 1a. Application of positive $V_g$ accumulates cations in the ion-gel electrolyte at the interface with the P-LSCO, producing a large electric field and removing $O^{2-}$ from the LSCO film (equivalently, introducing $V_O$s)[14], in a non-volatile fashion. It is known from prior work that this mechanism initially reduces P-LSCO (i.e., increases $\delta$), then leads to the formation of BM-LSCO in a matrix of P-LSCO (i.e., a P-BM mixed phase), and then finally to pure BM-LSCO, as a function of $V_g$[14]. The P and BM structures are shown schematically in Fig. 1b, where the perovskite phase (left) is fully oxygenated (with $CoO_6$ octahedra), while the BM phase (right) consists of perovskite layers separated by comparatively oxygen-deficient $CoO_4$ tetrahedra[10–12,14,29]. The $V_O$s in BM are ordered not only in alternate planes along the c-axis, but also in lines in the a-b-plane. The P → BM transformation thus quadruples the c-axis lattice parameter of the original P-LSCO and lowers the lattice symmetry from cubic to orthorhombic[10–12,14,29].

Figure 1c shows high-resolution wide-angle specular XRD scans of representative LSCO film devices after ion-gel gating to various $V_g$ from 0 to 3.5 V. Ion-gel gating was performed by sweeping $V_g$ at 0.5 mV s$^{-1}$ to the target voltage. Two-wire transport measurements

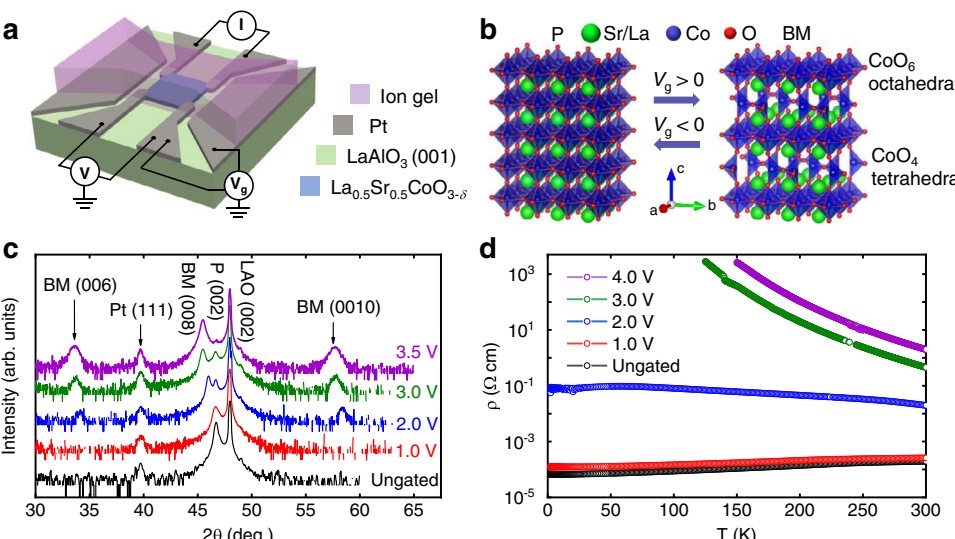

**Fig. 1 | Electrolyte gating and the P → BM phase transformation. a** Schematic of the $La_{0.5}Sr_{0.5}CoO_{3-\delta}$-film-based side-gated electrolyte-gate devices used in this study, with an overlying ion gel as the electrolyte. I and V are the current and voltage for measurement of four-wire channel resistance, while $V_g$ is the gate voltage. **b** Crystal structure of P-LSCO (left) and BM-LSCO (right). When $V_g$ is applied, the LSCO film is transformed from P → BM and vice-versa depending on the $V_g$

polarity: $V_g > 0$ drives to BM, and $V_g < 0$ drives to P. **c** X-ray diffraction from LSCO devices on LAO(001) substrates after gating to various $V_g$. The y-axis represents the XRD intensity in arbitrary units, plotted on a logarithmic scale and offset for better visualization. **d** T-dependence of the resistivity (from a four-wire measurement) for selected LSCO films after gating to various $V_g$.

were made during the sweep, while four-wire $T$-dependent transport and TDTR measurements were made ex situ, after bias removal, enabled by the non-volatility of the gate effect[10–12,14,29]. Purely to facilitate TDTR measurements (which require the deposition of an overlying metallic transducer film), each scan here is from a different ion-gel-gated sample. The ungated (black) and +3.5 V (purple) scans were performed on larger-channel-area devices ($4 \times 3.5$ mm$^2$), specifically for better XRD signal-to-noise ratio (SNR), while the other (intermediate-voltage) scans (1.0 V, 2.0 V, and 3.0 V) were performed on $1 \times 1$ mm$^2$ channel devices, as used for four-wire transport measurements. The ungated data (black) reveal the expected phase-pure epitaxial P-LSCO on the LAO(001) substrate (in addition to a Pt(111) peak from the device electrodes), consistent with previous reports on LSCO deposited via high-pressure-oxygen sputtering[6,14–16,30]. Additional characterization of P-LSCO films can be found in Supplementary Section 1 and Supplementary Fig. 1, establishing high epitaxial quality and full strain to the substrate.

Moving to the gated-LSCO film XRD patterns in Fig. 1c, we find a clear downshift of the P(002) peaks with increasing $V_g$, indicating expansion of the P-LSCO $c$-axis lattice constant due to $V_O$ formation in the P phase[14,15] (more information on $c$-axis lattice parameters vs. $V_g$ can be found in Supplementary Table 1). At 2 V and beyond, a (001)-oriented BM structure is then detected via the emergence of additional peaks around 34 and 58°, due to the BM(006) and BM(0010) reflections. This signifies the quadrupling of the unit cell in the $V_O$-ordered BM phase. Initially, these BM-LSCO peaks occur along with P(002) peaks, indicating a mixed phase region, as observed previously and interpreted in terms of phase coexistence across a first-order P → BM transformation[14]. At 3 V, however, the P(002) peak continues to downshift, indicating further lattice expansion, additional $V_O$ formation, and near-complete transformation to BM (the P(002) reflection then becomes the BM(008)). Concomitantly, the BM(006) and (0010) peaks intensify, while the P(002) peak diminishes. By 3 V, the intensity ratio of the BM(008) to BM(006) reflections is in fact ~70, relatively close to the value of ~40 previously observed for fully transformed BM-LSCO after ion-gel gating[14]. Interestingly, little difference is then observed between films gated to 3 and 3.5 V. We interpret this as resistance to further removal of O$^{2-}$ beyond the BM-LSCO stoichiometry (i.e., $\delta$ ~ 0.5) at voltages above 3 V. Regarding the structural perfection of the gated BM-LSCO, more detailed analysis (peak fitting) of the 3.5 V sample (purple) on the larger-channel-area-device revealed additional disorder in the $V_O$ sublattice compared to as-deposited BM films (e.g., SrCoO$_{2.5}$) reported in literature[11,12,31–34] (see Supplementary Section 1). It should also be noted that the P(002) peak intensity, although weak, still remains after 3 V. This relatively low-intensity peak arises from the portion of the P-LSCO film buried beneath the Pt contacts, however, which is ungated. Overall, we thus conclude from Fig. 1c that a reduced P phase exists up to ~1 V, beyond which P + BM phase coexistence persists to ~3 V, followed by near phase-pure BM. These findings are in good general agreement with prior LSCO gating studies[14]. More detailed information on these gated LSCO films, including thicknesses, applied $V_g$, and estimated phases can be found in Supplementary Section 1. As a final comment on these data, we note that, consistent with prior work on gated LSCO[14,15], we find no significant evidence of the hydrogenated phase seen in gating studies of SrCoO$_{3-\delta}$. The transformation in this work is thus simply between P and BM phases; it is therefore bi-state not tri-state switching.

## Electronic transport properties of ion-gel-gated LSCO films

Due to the very different electronic characteristics of the LSCO P and BM phases, major changes in electronic transport accompany the $V_g$-induced P → P + BM → BM transformation deduced above. Figure 1d shows electrical resistivity ($\rho$) vs. temperature ($T$) data for an ungated LSCO film, as well as films gated to 1–4 V. The ungated LSCO film is a metallic ferromagnet with a residual resistivity $\rho_0 \approx 70$ μΩ cm, a

residual-resistivity-ratio of 2.95, and $T_C \approx 220$ K. This is as expected for relatively thick films of La$_{1-x}$Sr$_x$CoO$_{3-\delta}$ with $x = 0.5$, where metallic conduction and long-range ferromagnetism are observed for $x > 0.17$[35–37]. After gating to 1 V, the metallic conduction in the LSCO film is maintained, albeit with slightly higher $\rho$, reflecting the additional $V_O$ density introduced into the P lattice. As electron donors, $V_O$ compensate holes (the dominant charge carriers in LSCO), thus increasing $\rho$. The relatively small change in $\rho$ for LSCO at 1 V compared to ungated LSCO is consistent with Fig. 1c in that the P phase is retained, but with a higher $\delta$. After applying 2 V, however, the RT $\rho$ increases nearly 100-fold, reflecting not only a substantial increase in $V_O$ concentration (larger $\delta$), but also entry into the P + BM phase coexistence region. The unusual form of $\rho$ vs. $T$ for the 2-V-gated sample (note the flattening at low $T$) is also consistent with this; lateral inhomogeneity would be expected in this situation and indeed we detected in-plane anisotropy in the orthogonal electrical resistances in four-wire van der Pauw measurements at low $T$. For the samples gated to 3 and 4 V, where XRD reveals near-phase-pure BM (Fig. 1c), the LSCO films then exhibit completely insulating behavior in Fig. 1d, with $\rho$(RT) of ~0.5 and ~2 Ω cm, respectively. This represents a factor of ~10$^4$ between the RT $\rho$ of ungated P-LSCO and BM-LSCO, comparable to literature values for electrolyte-gated SrCoO$_{3-\delta}$ and La$_{0.5}$Sr$_{0.5}$CoO$_{3-\delta}$[12,14].

In addition to these ex situ measurements of $T$-dependent four-wire resistivity (Fig. 1d), in-situ measurements of two-terminal resistance (i.e., source-drain resistance) were also made while sweeping to the target $V_g$ for each sample (see Supplementary Fig. 2). These two-terminal measurements show reassuring overall agreement with the ex-situ trends in resistivity depicted in Fig. 1d, i.e., minor increases in resistance before ~2 V, a sharp increase between ~2–3 V, then a region with relatively weak $V_g$ dependence, indicating completion of the P → P + BM → BM transformation. These data (Supplementary Fig. 2) also highlight the high reproducibility of the gating process, underscoring the robustness of our approach.

## Tuning of LSCO thermal conductivity with gate voltage

The thermal conductivities of LSCO thin films gated to various $V_g$ (from 0 to 4 V) were measured with the TDTR approach, which is an ultrafast-laser-based pump-probe technique[38,39]. All TDTR measurements were made ex-situ, after biasing and ion-gel removal, taking advantage of the non-volatile nature of the electrolyte gating in this case[10–12,14,29]. Figure 2a illustrates the sample configuration for TDTR measurements, which consists of a thin layer of Pt deposited on top of the LSCO films, serving as the heat source and thermometer. Pt was chosen as the transducer for this study, instead of the typical Al, specifically to avoid Al-gettering-induced $V_O$ formation in the underlying LSCO[40]. By varying the modulation frequency in the TDTR approach, the measurement sensitivity to the LSCO thermal conductivity and the thermal interfaces can be tailored. This enables us to also extract the interfacial

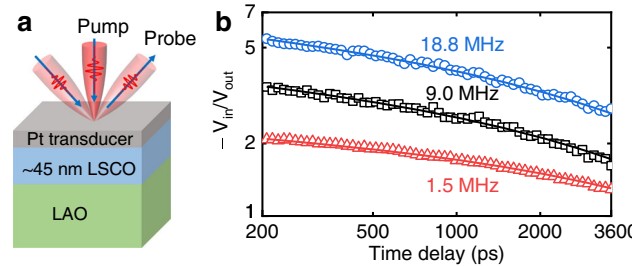

**Fig. 2 | Thermal measurements using the TDTR approach. a** Sample configuration for TDTR measurements. **b** Representative TDTR ratio signals of the ungated LSCO film measured at three modulation frequencies. The solid lines are the best fits to the through-plane thermal measurement data based on a heat diffusion model. The data reduction gives a through-plane thermal conductivity of $4.6 \pm 2.0$ W m$^{-1}$ K$^{-1}$ for this ungated LSCO film.

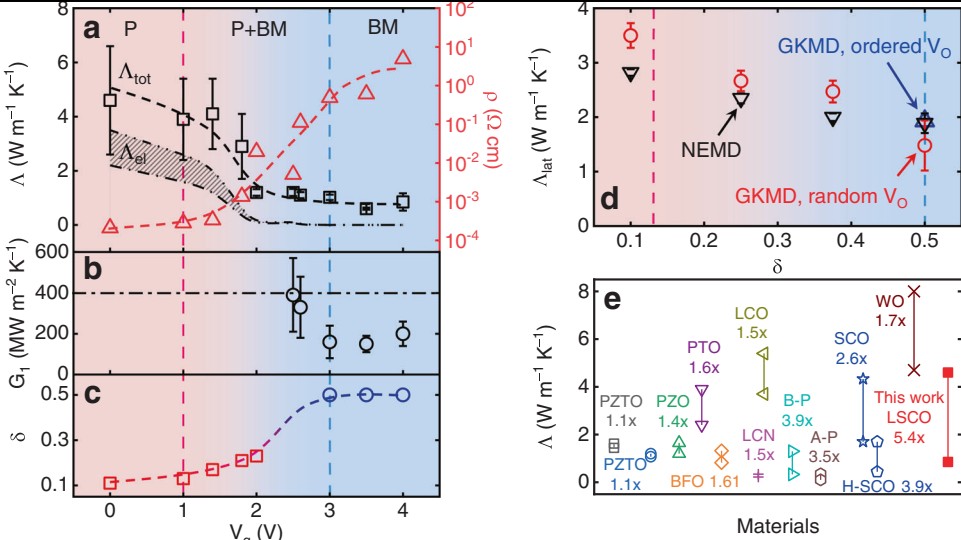

**Fig. 3 | Impact of $V_g$ on thermal properties. a** The thermal conductivity (left axis) and electrical resistivity (right axis) of LSCO films, **b** the interfacial thermal conductance between Pt and LSCO, and **c** oxygen non-stoichiometry due to vacancies, as functions of $V_g$. In panel **a**, $\Lambda_{tot}$ represents the measured total thermal conductivity of the LSCO films. The measurement uncertainty of LSCO thermal conductivity decreases versus $V_g$ from ~40% to ~10%. $\Lambda_{el}$ represents the electronic thermal conductivity estimated from electrical conductivity based on the Wiedemann-Franz law. The black dash-dotted lines represent the upper and lower limits of $\Lambda_{el}$ based on different Lorenz numbers ($L$). The dashed lines serve as guides to the eye. In panel **b**, the measurement uncertainty shown in the figure also decreases versus $V_g$, from ~50% to ~30%. In panel **c**, $\delta = 0.5$ (blue circles) is assumed for the complete transition to brownmillerite ($V_g \geq 3$ V), as our method used to estimate $\delta$ (red squares) is not valid above $\delta \approx 0.25$ (see Methods). The red-blue graded dashed line serves as a guide to the eye. **d** MD simulation results, obtained from both Green-Kubo MD (GKMD) and nonequilibrium MD (NEMD), for the lattice thermal conductivity of LSCO with different $\delta$. The uncertainty of calculation results is within 10%. In all panels, the vertical pink and blue dashed lines represent, respectively, the starting and ending points of the P → P + BM → BM transformations. **e** Comparison of thermal conductivity tuning factor through bi-state tuning process in this work and previous experimental studies[10,19–28], including PbZr$_{0.3}$Ti$_{0.7}$O$_3$ (PZTO, 5 cycles)[20,21], PbZrO$_3$ (PZO, 10+ cycles)[22], PbTiO$_3$ (PTO)[23], BiFeO$_3$ (BFO)[24], bio-polymers (B-P, 10+ cycles)[25], azobenzene polymers (A-P, 6 cycles)[26], liquid crystal networks (LCN)[27], Li$_x$CoO$_2$ (LCO, 2 cycles)[19], WO$_3$ (WO, 3 cycles)[28], and SrCoO$_{3-\delta}$ (SCO, 1 cycle, and H-SCO)[10]. Here, one cycle is defined as the transformation from state/phase A to state/phase B, and then return to state/phase A. Cycle numbers are provided when available from the literature. Note: During the review period of this work, several latest papers appeared reporting the tuning of materials' thermal conductivities that were not included in Fig. 3e. These latest studies report a tuning ratio of ~4x for the thermal conductivity of SCO[70], -1.27x for the thermal conductivity of ferroelectric Pb(Mg$_{1/3}$Nb$_{2/3}$)O$_3$–PbTiO$_3$[71], and -1.05x for the thermal conductivity of PZT[72] at near room temperature.

thermal conductance between the Pt transducer film and the LSCO film ($G_1$), and between the LSCO film and the LAO substrate ($G_2$), by combining data reduction of routine TDTR measurements with dual-frequency TDTR analyses (see Supplementary Section 3 for details).

As part of the input parameter set used in the analysis of TDTR data to determine the thermal conductivity of LSCO films, the thickness ($d_{Pt}$) and thermal conductivity ($\Lambda_{Pt}$) of the Pt transducer were obtained, respectively, from grazing incidence X-ray reflectometry and four-point probe measurements (incorporated with the Wiedemann-Franz law, WFL) of a standard Si/SiO$_2$(300 nm)/Pt(~70 nm) reference sample at RT. For TDTR data reduction as a function of $T$, $\Lambda_{Pt}$ is set based on literature data[41], as is the $T$-dependent volumetric heat capacity of the transducer ($C_{Pt}$)[42]. In addition to the Pt transducer, some dimensional parameters and thermal properties of the LSCO film ($d_{LSCO}$, $C_{LSCO}$) and LAO substrate ($\Lambda_{LAO}$, $C_{LAO}$) are also needed. The $T$-dependent $\Lambda_{LAO}$ data were obtained from TDTR measurements of a bare LAO substrate. $C_{LSCO}$ and $C_{LAO}$ were calculated as a function of $T$ based on the Debye model using the literature reported Debye temperature[43,44], and are in good agreement with literature data[43,45–48] (see details in Supplementary Section 6 and Supplementary Fig. 7). For TDTR measurements at low $T$ ($T < 273$ K) where the material's thermal properties (i.e., specific heat and thermal conductivity) exhibit stronger $T$ dependence, we also performed $T$-dependent corrections, taking into account the impact of steady-state heating and the pump per-pulse heating (details are presented in Methods and Supplementary Section 6).

Figure 2b shows representative TDTR ratio data of the in-phase and out-of-phase thermoreflectance signals ($-V_{in}/V_{out}$) measured on the ungated LSCO film at RT. Three modulation frequencies (1.5, 9.0, and 18.8 MHz) were used to enable tailoring of the measurement sensitivity. The solid lines in Fig. 2b are best fits to the TDTR data based on a heat diffusion model for a multi-layer structure[38,49]. At RT, the fitted through-plane thermal conductivity of the ungated LSCO sample is $4.6 \pm 2.0$ W m$^{-1}$ K$^{-1}$. The interfacial thermal conductances of the Pt/LSCO ($G_1$) and LSCO/LAO ($G_2$) interfaces are determined as 400 and 800 MW m$^{-2}$ K$^{-1}$, respectively. More details about the TDTR thermal measurements and the extraction of $G_1$ and $G_2$ are provided in Methods, and Supplementary Section 3.

The RT electrical resistivity and thermal conductivity of LSCO thin films are summarized as functions of $V_g$ in Fig. 3a. As $V_g$ increases from 0 to 4 V, the change in $\rho$ reaches almost five orders of magnitude, a clear reflection of the metal-insulator transition shown in Fig. 1d. Coincident with this, the central experimental result of this work is that the through-plane thermal conductivity of LSCO decreases from $4.6 \pm 2.0$ to $0.85 \pm 0.32$ W m$^{-1}$ K$^{-1}$ as $V_g$ varies from 0 to 4 V, i.e., a factor of more than 5 reduction via ion-gel gating. We consider this record-high tuning factor to be for the through-plane thermal conductivity, acknowledging the possibility of thermal transport anisotropy in the BM-LSCO film. It is reasonable, however, to convert the in-plane electrical resistivity to a through-plane thermal conductivity for the purposes of comparison, as we discussed in Supplementary Section 7. We emphasize that the thermal conductivity can be continuously tuned over this range, in a non-volatile fashion, at room temperature. It should be noted that the thermal conductivity of our as-deposited P-LSCO films (with $x = 0.5$ and $\delta \approx 0.1$, see Methods and the discussion below regarding the determination of $\delta$) is $4.6 \pm 2.0$ W m$^{-1}$ K$^{-1}$, which is less than half the equivalent values for several unsubstituted ABO$_3$ perovskites (e.g., 11 W m$^{-1}$ K$^{-1}$ for SrTiO$_3$[50], 13 W m$^{-1}$ K$^{-1}$ for BaSnO$_3$[51], and 13 W m$^{-1}$ K$^{-1}$ for LaAlO$_3$[52]). Also, prior experimental studies reported a bulk thermal conductivity of 6 W m$^{-1}$ K$^{-1}$ for single-crystal

$La_{0.7}Sr_{0.3}CoO_3$[52,53]. The significantly smaller thermal conductivities of $La_{0.7}Sr_{0.3}CoO_3$ and $La_{0.5}Sr_{0.5}CoO_3$ are partially attributed to enhanced phonon-defect scattering resulting from the mass mismatch and local strains induced by Sr substitution for La[51-53]. Below, we analyze the change of thermal conductivity versus $V_g$ in detail, including discussing the contributions to the thermal conductivity from various critical factors.

It is important to note, however, that the $V_g$-driven P → BM and metal-insulator transitions also impact thermal transport across the Pt/LSCO interfaces. Figure 3b summarizes the results on $G_1$ extracted from TDTR, where a smaller $G_1$ is observed when LSCO is gated to higher $V_g$. For BM-LSCO, with insulating behavior, heat can be transferred across the Pt/BM-LSCO interface via two channels: electron-phonon interactions in Pt in series with the phonon–phonon interactions (between Pt and BM-LSCO phonons). The average measured $G_1$ (170 MW m$^{-2}$ K$^{-1}$) of our BM-LSCO is comparable to the values of interfacial thermal conductance between Pt and oxides reported in literature[54-57] (Supplementary Fig. 4). However, for $V_g$ < 3 V, the P phase in the LSCO films provides an additional channel for interfacial heat transfer, via electron-electron interactions (between Pt and P-LSCO) across the interface[58,59], thus leading to larger $G_1$. This increase in $G_1$ reduces the measurement sensitivity to $G_1$ (Supplementary Fig. 3). As a result, we used a nominal value of $G_1$ (400 MW m$^{-2}$ K$^{-1}$, black dashed line in Fig. 3b) for LSCO films at $V_g$ < 2 V, when measurement sensitivity is insufficient to determine $G_1$ (i.e., the choice of $G_1$ value does not change the fitting results of the LSCO thermal conductivity). Besides, $G_2$ of the LSCO/LAO epitaxial interface is measured to be 800 WM m$^{-2}$ K$^{-1}$ using the dual-frequency TDTR approach with enhanced measurement sensitivity to $G_2$[60] (see Supplementary Section 3 for more details). This value of $G_2$ is consistent with literature data for strongly bonded interfaces[61].

## Effects of the P → BM transformation and metal-insulator transition on the thermal conductivity of LSCO

As already noted, the $V_g$-driven P → BM phase transformation in electrolyte-gated LSCO involves several factors, including a large change in $\delta$, lowered crystal symmetry, $V_O$ ordering, and an accompanying metal-insulator transition. The P → BM (cubic to orthorhombic) crystal structure change (Fig. 1b, c), induced by the $V_g$-driven increase in $\delta$, impacts phonon transport, thus modifying thermal conductivity via the phonon contribution. To better quantify the impact of $V_O$ on the thermal conductivity in Fig. 3a, it is useful to correlate $V_g$ with $V_O$ concentration. To this end, we estimated $\delta$ using an established empirical approach (see Methods), based on electrical transport data[14,16,52]. The resulting values are plotted in Fig. 3c. The expected increasing trend in $\delta$ is observed for $V_g$ from 0 to 2 V. For films gated above 2 V, however, our approach is no longer applicable (see Methods), and we simply assign the nominal $\delta = 0.5$ implied by our XRD observation of near-phase-pure BM-LSCO at 3, 3.5, and 4 V (blue circles in Fig. 3c).

To gain further insight into the thermal conductivity change in Fig. 3a, we decompose it into electronic and lattice contributions, i.e., $\Lambda_{tot} = \Lambda_{el} + \Lambda_{lat}$ (with $\Lambda_{el}$ being the electronic thermal conductivity and $\Lambda_{lat}$ the lattice thermal conductivity). $\Lambda_{el}$ can be estimated from the electrical conductivity ($\sigma = 1/\rho$) via the WFL, i.e., $\Lambda_{el} = L\sigma T$, with $L$ being the Lorenz number. Since the value of $L$ varies non-trivially depending on the nature of the material system, we used the lower and upper limits on $L$ commonly accepted in the literature (1.5 and 2.44 × $10^{-8}$ V$^2$ K$^{-2}$), to estimate $\Lambda_{el}$[62-65]. The results are plotted as the shaded envelope between black dash-dotted lines in Fig. 3a, exhibiting the expected generally decreasing trend with $V_g$ due to the metal-insulator transition. For the as-deposited P-LSCO film with $\delta \approx 0.1$ (Fig. 3c), the lower and upper limits of $\Lambda_{el}$ are estimated as 2.2 and 3.5 W m$^{-1}$ K$^{-1}$, resulting in the upper and lower limits of $\Lambda_{lat} = \Lambda_{tot} - \Lambda_{el}$ of 2.4 and 1.1 W m$^{-1}$ K$^{-1}$, respectively. The upper limit of $\Lambda_{lat}$ decreases from 2.4 to

0.85 W m$^{-1}$ K$^{-1}$, a 65% reduction, as $V_g$ increases from 0 to 4 V, at which point the P → BM transformation is complete. Importantly, we thus find that electrons contribute comparably with phonons for P-LSCO, which is reasonable for "dirty", i.e., higher resistivity, metals[66]. This is notably different from the P-SrCoO$_{3-\delta}$ reported by Lu et al.[10], where the electronic contribution to thermal conductivity was deduced to be negligible. We attribute this difference to the likely larger $\delta$ and disorder level in the P phase in ref. 10, where the P-SrCoO$_{3-\delta}$ was achieved from as-deposited BM films via gating, in contrast to our as-deposited high-epitaxial-quality P-LSCO. Supporting this, Lu et al. reported $\rho$(RT) ≈ 5 × $10^{-2}$ Ω cm in their P-SrCoO$_3$[10], compared to $\rho$(RT) ≈ 2 × $10^{-4}$ Ω cm in our as-deposited P-LSCO, i.e., our P conductivity is ~250 times higher.

To further understand the trend of $\Lambda_{lat}$ versus $V_g$ (and therefore also $\delta$), we performed Green-Kubo MD (GKMD) and nonequilibrium MD (NEMD) simulations of thermal conductivity (see Supplementary Section 5 for calculation details), with results shown in Fig. 3d. The GKMD and NEMD results agree reasonably well. The MD simulations predict a ~62% $\Lambda_{lat}$ reduction (from 3.9 to 1.5 W m$^{-1}$ K$^{-1}$) as $\delta$ increases from 0.1 to 0.5, assuming random spatial distributions of $V_O$s. Considering that the $V_O$s in the BM phase of our LSCO films are actually ordered[52], we also calculated the thermal conductivity of orthorhombic BM-LSCO, with ordered $V_O$s, using MD simulations; this yields a larger $\Lambda_{lat} = 1.9$ W m$^{-1}$ K$^{-1}$ at $\delta = 0.5$ due to the decrease of phonon-defect scattering. There is thus a ~50% lattice thermal conductivity reduction from $\delta = 0.1$ in the disordered P phase to $\delta = 0.5$ in the ordered BM phase. Importantly, these 62% and 50% reductions in $\Lambda_{lat}$ agree reasonably with the upper limit of 65% obtained from our $\Lambda_{lat} = \Lambda_{tot} - \Lambda_{el}$ extraction. Note that the absolute values of $\Lambda_{lat}$ from the experiment and MD simulations are not directly compared because the classical interatomic potential used in our MD simulations provides only qualitative insights. Based on the above discussion, we conclude that the decrease in thermal conductivity with increasing $V_g$ originates from comparable reductions in both the electronic and lattice contributions. This plays an important role in our record-high (>5-fold) tuning of thermal conductivity achieved here via bi-state gating. Figure 3e summarizes the tuning factor of material thermal conductivities in our work, and in prior experimental studies, via a bi-state tuning process[10,19-27]. Evidently, the tuning factor of LSCO thermal conductivity in our work is the largest among experimental reports.

Considering the reversible nature of the $V_g$-induced P ↔ BM phase transformation in LSCO, we also reverted some gated BM-LSCO films to the P phase by applying reverse gate voltages of up to −4.5 V at RT. While this typically generates an approximately $10^4$ decrease in RT $\rho$ due to the BM → P insulator-metal transition, the resulting RT $\rho$ values of recovered P-LSCO films are typically double that of the initial as-deposited P films, corresponding to approximately half the initial electronic thermal conductivity. The thermal measurement of one such film generated a recovered P-LSCO thermal conductivity of 3.5 ± 1.1 W m$^{-1}$ K$^{-1}$ at RT. This ~20% reduction in the $\Lambda_{tot}$ of recovered P-LSCO compared to the as-deposited P-LSCO film no doubt arises from additional structural disorder induced during the cyclic gating (see Supplementary Section 8 for more discussion). Nevertheless, such data confirm the overall reversibility of the approach presented here. Further improvement of reversibility could be achieved through better device design and optimized gating conditions.

## Temperature-dependent thermal conductivities of LSCO

To further clarify the mechanisms of the scattering processes of the relevant heat carriers in LSCO, we also performed $T$-dependent thermal measurements on P-LSCO (ungated) and BM-LSCO ($V_g$ = 3 V) from ~90 to ~500 K. Generally, as $T$ increases, the thermal conductivity of a dielectric crystal should first increase (due to the increase of the specific heat)[67] and then decrease with a $1/T$ trend (due to the increase of Umklapp scattering)[68]. However, neither of our LSCO samples exhibit such typical features. Instead, both P-LSCO and BM-LSCO show a

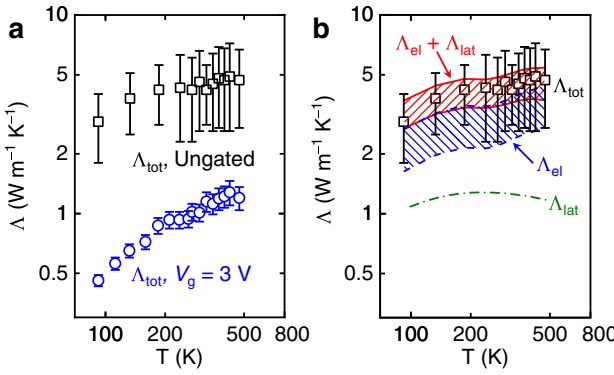

**Fig. 4 | *T*-dependent thermal conductivity. a** *T*-dependent thermal conductivities of P-LSCO (ungated, black squares) and BM-LSCO ($V_g = 3$ V, blue circles). The measurement uncertainty for P-LSCO is around 40% while for BM-LSCO it is around 10%. **b** *T*-dependent $\Lambda_{el}$ and $\Lambda_{lat}$ of P-LSCO. Boltzmann transport equation (BTE) calculations were performed to predict $\Lambda_{lat}$ for a 45-nm La$_{0.5}$Sr$_{0.5}$CoO$_{3-\delta}$ ($\delta = 0.1$) film (green dashed-dotted line), while $\Lambda_{el}$ (blue dashed line) was calculated from the temperature dependence of the resistivity (Fig. 1d) using the WFL with the common lower and upper limits of $L$. Red solid lines represent the sum of $\Lambda_{el}$ and $\Lambda_{lat}$.

monotonically increasing trend with *T* in our measurement range (Fig. 4a). We interpret this observation in terms of the two relevant types of heat carriers (electrons and phonons), with differing *T* dependencies. In P-LSCO, clearly from the above, both electrons and lattice vibrations contribute to thermal transport. The $\Lambda_{el}$ estimated from the WFL with the lower and upper limits of $L$ (using our measured $\rho(T)$) is presented as the envelope shaded with blue dashed lines in Fig. 4b. It can be seen that $\Lambda_{el}$ increases with *T*, dominating the increase in $\Lambda_{tot}$. It is worth noting that a dip is observed in $\Lambda_{el}$ at ~235 K, however. This is because the Curie temperature of our P-LSCO films is ~230 K[14,37], at which the transition from ferromagnetic to paramagnetic behavior induces a change in the $\rho(T)$ behavior, as can be seen in Fig. 1d upon close inspection.

The *T*-dependent $\Lambda_{lat}$ of P-LSCO was also calculated using the BTE to take into account the size effect in the thin film, and is shown as the green dash-dotted line in Fig. 4b[52]. $\Lambda_{lat}$ exhibits a peak at around 200 K. Generally, the peak temperature of thermal conductivity of insulating single-crystal perovskite oxides in unsubstituted ABO$_3$ perovskites appears around or below 100 K (including LaAlO$_3$, the substrate used in this work, see Supplemental section 6)[43,45–48]. The difference in peak temperature between our P-LSCO and other such ABO$_3$ perovskites arises because of the presence of a significant V$_O$ concentration, the disorder on the A-site (La/Sr), and the sample boundaries, inducing strong *T*-independent extrinsic phonon scattering, leading to a higher peak temperature in $\Lambda_{lat}$ compared to pristine ABO$_3$ crystals. Finally, we summed $\Lambda_{el}$ and $\Lambda_{lat}$ ($\Lambda_{el} + \Lambda_{lat}$, shaded area with red solid lines in Fig. 4b) to compare with $\Lambda_{tot}$ obtained from measurements, obtaining a reasonably good agreement.

In contrast to P-LSCO, BM-LSCO is strongly insulating, with lattice vibrations therefore being the dominant heat carriers, and thus $\Lambda_{tot} \approx \Lambda_{lat}$. However, $\Lambda_{lat}$ of BM-LSCO still does not show a peak in the probed temperature range, in contrast to typical behavior of pristine insulating crystals. The reason for this is that BM-LSCO contains an even larger density of V$_O$s, a lattice structure with a lower symmetry due to those V$_O$s, and additional disorder associated with the A-site cations (La/Sr), resulting in strong extrinsic phonon scattering, even at low *T*. Therefore, the thermal transport in BM-LSCO is more glass-like, rather than crystal-like. Supplementary Fig. 8 also shows the *T*-dependent thermal conductivity of one BM-LSCO sample gated to 4 V, which exhibits a similar trend to that of the BM-LSCO sample at 3 V shown in Fig. 4a, further supporting these results and conclusions.

## Discussion

In summary, we have demonstrated that by applying single-step solid-state electrolyte gating at RT we can achieve a record-high (>5-fold) tuning of the thermal conductivity of ion-gel-gated LSCO films in a non-volatile, continuous manner. This large tunability in the thermal conductivity of epitaxial LSCO films results from the structural modifications and metal-insulator transition induced by the P → BM phase transformation. As $V_g$ increases, more oxygen vacancies are created and the lattice symmetry is lowered, leading to a sizable reduction in the LSCO lattice thermal conductivity. Accompanying these structural modifications, the metal-insulator transition across the P → BM transformation further extends the extent of the tuning of thermal conductivity by controlling the electronic contributions to the thermal conductivity. In addition, *T*-dependent thermal conductivity measurements provided further physical insights into the scattering mechanisms of the relevant heat carriers. For metallic P-LSCO, the observed increasing trend of $\Lambda_{tot}$ *vs.* *T* is partially attributed to the substantial electronic contributions to thermal transport, which are absent for BM-LSCO. For insulating BM-LSCO, the increase in thermal conductivity with *T* instead represents more of the trend typical of amorphous oxides due to the extensive structural defects (e.g., higher V$_O$ concentrations and A-site disorder). This success in actively tailoring the thermal conductivity of thin-film LSCO by over a factor of five, via electrolyte gating, opens the possibility of voltage-driven tunable thermal materials for applications in thermal management and energy conversion, which require dynamic control of heat propagation.

## Methods

### LSCO film growth, device fabrication, and electrolyte gating

Thin films of LSCO were deposited on commercial (MTI corp.) 5 × 5 mm$^2$ and 10 × 10 mm$^2$ LaAlO$_3$(001) substrates using high-pressure-oxygen sputtering, from 2" polycrystalline sputtering targets of the same nominal stoichiometry. Substrates were first annealed at 900 °C in flowing O$_2$ (99.998%, ~1 Torr) for 15 min, before being cooled to 600 °C for deposition. LSCO was then deposited via DC sputtering at 600 °C in flowing O$_2$ (1.5 Torr) at a power of 60–70 W. This produced epitaxial films at a deposition rate of ~20 Å/min. After deposition, films were cooled to RT in 600 Torr of O$_2$ at ~15 °C/min. These procedures essentially employ optimized parameters reported in prior studies[6,14–16,30,37]. The thicknesses of films in this work varied from 45 to 58 nm, estimated from established growth rates and cross-checked from wide-angle XRD Laue fringe spacings.

Sputtered LSCO films were then used to fabricate electrolyte-gate transistor devices[6,14,15]. As-deposited LSCO films were first Ar-ion milled selectively using steel masks to form LSCO channels of 1 × 1 mm$^2$. Subsequent Mg(5 nm)/Pt(50 nm) gate and contact electrodes were then sputter-deposited through a separate mask and annealed in O$_2$ (450 °C, 5 min). To complete side-gated transistors, a single piece of ion gel was then cut and laminated atop the LSCO channel, contact electrodes (partially), and gate electrodes (substantially). A schematic of the final device is shown in Fig. 1a. The ion gels were fabricated by spin coating (to ~50 μm thickness) a solution of ionic liquid and polymer onto ~1 inch$^2$ glass wafers. The ionic liquid solution consisted of: 1-ethyl-3-methylimidazolium bis(trifluoro-methylsulfonyl) imide (EMI-TFSI) ionic liquid, poly(vinylidene fluoride-cohexafluoropropylene) polymer, and acetone, in a weight ratio of 1:4:8, respectively. Gating was performed at RT, in vacuum (<1 × 10$^{-5}$ Torr), sweeping $V_g$ at 0.5 mV s$^{-1}$ to the specified target values; $V_g$ was applied between the side gate pads and two diagonal electrodes shorted to the film channel (Fig. 1a). During $V_g$ sweeps, two-wire resistance measurements were made in situ between two diagonal electrodes (the two contacts that were not shorted to the gate counter-electrode), representing a source and drain. Such measurements were made by applying a voltage ($V_{SD}$) of ±0.2 V and measuring a current ($I_{SD}$) (see

Supplementary Fig. 2). $V_g$ and $V_{SD}$ were applied with separate Keithley 2400 (K2400) source-measure units, while the gate current ($I_g$) and $I_{SD}$ were measured with the corresponding K2400 units. After reaching the desired $V_g$, gating was terminated by disconnecting the voltage supply to the gates, removing the ion gel, and cleaning the film surface with acetone to remove any residual ion gel.

### X-ray diffraction

LSCO film devices of two dimensions ($1 \times 1$ mm$^2$ films on $5 \times 5$ mm$^2$ substrates, and $4 \times 3.5$ mm$^2$ films on $10 \times 10$ mm$^2$ substrates) were characterized both before and after gating via high-resolution specular wide-angle XRD (using a Rigaku Smartlab XE, with Cu Kα radiation). Reciprocal space mapping was performed with the same instrument and settings. Synchrotron XRD measurements (shown only in Supplementary Fig. 1a) were performed on a representative $4 \times 3.5$ mm$^2$ LSCO film deposited on a $10 \times 10$ mm$^2$ LAO(001) substrate at the 12-ID-D beamline of the Advanced Photon Source at Argonne National Lab. The synchrotron XRD set-up was equipped with a six-circle Huber goniometer and a Pilatus II 100 K area detector. The spot size and X-ray beam energy were ~500 $\mu$m and 22 keV (λ - 0.56 Å), respectively. The measurements were carried out at a temperature of 150 K with a liquid-N$_2$ gas flow cryocooler.

### Electronic transport

Four-wire electronic transport measurements were taken in the van der Pauw geometry before and after electrolyte gating of LSCO films (ex situ), to determine the LSCO channel resistivity. Temperature-dependent data were acquired in a Quantum Design Physical Property Measurement System (PPMS). Electronic transport measurements were taken using either quasi-AC with the PPMS internal bridge (for metallic samples), or DC with a Keithley 2400 current source and a Keithley 2002 multimeter (for insulating samples).

### Determination of V$_O$ concentrations, $\delta$

The oxygen deficiency, $\delta$, of P-phase LSCO films was estimated using a previously established and validated method[14], in which the measured film resistivity is compared to La$_{1-x}$Sr$_x$CoO$_3$ ($x \leq 0.30$) single crystal resistivity data (where $\delta$ is close enough to zero to be considered negligible) to estimate the film $\delta$. First, the known low-temperature single crystal resistivity is plotted vs. $x$, then the measured film resistivity is used to interpolate an effective $x$ value, $x_{eff}$, from this master curve. Using $x_{eff} = x - 2\delta$ (i.e., assuming each V$_O$ dopes 2 electrons), with $x = 0.5$ in our case, the film $\delta$ can then be estimated. This estimation can be made only in the P phase, and only up to $\delta = 0.25$. We do apply this method in the mixed phase region, where the current is expected to be shunted by P regions, but it cannot, therefore, be applied in the BM phase. As noted in connection with Fig. 3c, at $V_g$ = 3, 3.5 and 4 V we simply assume $\delta \approx 0.5$ based on the BM structure evident from XRD.

### TDTR measurements

The thermal conductivities of LSCO films were measured with TDTR[49]. Prior to TDTR thermal measurements, a ~70 nm Pt layer was sputter-deposited onto the LSCO films (after gating to the desired $V_g$), to act as a transducer layer for TDTR measurements. At the laser wavelength of 783 nm, the extinction coefficient is ~7.4 for Pt[69]; therefore, the thickness of 70 nm is sufficient for the Pt transducer to be considered optically opaque. A Si/SiO$_2$ substrate was also coated with Pt in the same deposition, providing a reference to cross-check the electrical and thermal properties of the Pt. In TDTR, a mode-locked Ti:sapphire laser produces a train of optical pulses (~100 fs in duration) at a repetition rate of 80 MHz and a central wavelength of ~780 nm. The laser is divided into a pump beam and a probe beam with two orthogonal polarizations by a polarizing beam splitter. The pump beam is modulated by an electro-optical modulator, which heats the sample.

The probe beam is modulated by a mechanical chopper and detects the temperature response of the sample upon pump heating. An objective lens is used to focus the pump and probe beams onto the sample surface. A mechanical delay stage then varies the optical path of the pump beam, which produces a time delay of up to 4 ns between pump heating and probe sensing. The probe beam reflected from the sample is collected with a fast-response Si detector and then amplified by an RF lock-in amplifier for the data reduction.

We used a 5× objective lens with a $1/e^2$ beam spot size of ≈12 $\mu$m for all samples. For temperature-dependent TDTR measurements over the range of ~90 to ~500 K, the sample was mounted on a heating/cooling stage. Specifically, measurements at low temperatures (from ~90 K to RT) were done under vacuum, while measurements at elevated temperatures (from RT to ~500 K) were carried out in air. A MK2000 series temperature controller was used to provide temperature control. The set temperature of the controller was defined as the setting temperature. The laser power was optimized as a compromise between the signal-to-noise ratio and steady-state heating ($\Delta T_s$) for each set temperature. For measurements at low temperatures, due to the significant decrease in heat capacity, the temperature rise induced by the pump per-pulse can also be large ($\Delta T_p$). Therefore, we performed a temperature correction for low-temperature measurements, taking into account both $\Delta T_s$ and $\Delta T_p$. The detailed procedures for this temperature correction are provided in Supplementary Section 6.

### Reporting summary

Further information on research design is available in the Nature Portfolio Reporting Summary linked to this article.

## Data availability

All data needed to evaluate the conclusions in this paper are present in the article and the Supplementary Information. All raw data generated in this study and used for figures in both the main article and the Supplementary Information have been deposited in the DRUM database with free access [https://doi.org/10.13020/kjhp-az69].

## Code availability

The code used to perform MD simulations is an open-source package (LAMMPS). All other in-house codes related to this work are available from the authors upon reasonable request.

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

## Acknowledgements

This work was primarily supported by the National Science Foundation (NSF) through the UMN MRSEC under DMR-2011401 (Y.Z., W.M.P., C.Z., V.C., K.H., C.L., and X.W.). Parts of this work were carried out at the Characterization Facility, UMN, which receives partial support from the NSF through the MRSEC program. Portions of this work were also conducted in the Minnesota Nano Center, which is supported by the NSF through the National Nanotechnology Coordinated Infrastructure under ECCS-2025124 (Y.Z., W.M.P., C.Z., V.C., K.H., C.L., and X.W.). This research used resources of the Advanced Photon Source, a U.S. Department of Energy (DOE) Office of Science user facility operated for the DOE Office of Science by Argonne National Laboratory under Contract No. DE-AC02-06CH11357 (Hua Z.). The MD and BTE calculations were supported by NSF under CBET-2212830 (R.X., Hao Z., and T.F.). The computation used resources of the National Energy Research Scientific Computing Center, supported by the Office of Science of the DOE under Contract DE-AC02-05CH11231 (R.X., Hao Z., and T.F.), the Center for High Performance Computing at the University of Utah, and the Extreme Science and Engineering Discovery Environment.

## Author contributions

C.L. and X.W. originated and supervised the research. Y.Z. and C.Z. carried out the TDTR measurements and analyses under the guidance of X.W. W.M.P., V.C., and K.H. prepared the LSCO samples under the guidance of C.L. W.M.P., V.C., and Hua Z. performed the XRD characterization. W.M.P. fabricated the devices and performed the electrolyte gating and electrical transport measurements and analyses, under the guidance of C.L. R.X., Hao Z., and T.F. did the MD simulations and BTE calculations. Y.Z., W.M.P., R.X., T.F., C.L., and X.W. contributed to the writing of the manuscript. All authors discussed the data and provided input on the paper.

## Competing interests

The authors declare no competing interests.
