## [Peer Review File · Nature Communications]

Wide-range continuous tuning of the thermal conductivity of La_{0.5}Sr_{0.5}CoO_{3-δ} films via room-temperature ion-gel gatingREVIEWER COMMENTS

Reviewer #1 (Remarks to the Author):

In this paper, the authors report tunable thermal conductivity in epitaxial $\text{La}_{0.5}\text{Sr}_{0.5}\text{CoO}_{3-\delta}$ (LSCO) films under ion gel gating. Using a combination of XRD, electrical transport and TDTR measurements, they show that the topotactic phase transition from perovskite and brownmillerite phase, driven by the deintercalation of oxygen ions via electrolyte gating causes a decrease in cross-plane thermal conductivity, and claim to realize ~5x modulation in the conductivity.

The main idea demonstrated in this work is utilizing the migration of ions driven by electrolyte gating to dynamically control the thermal properties of oxide thin films, which, however, has been already reported in a couple of recent studies. Furthermore, the modulation by a factor of 5 seems questionable. As the authors mentioned in Line 347 on Page 17, after gating the BM-LSCO back to the P phase using -4.5 V, the thermal conductivity only increases to 2.7 from 0.85 W/(m K), meaning a tuning factor of ~3. Also, it is unclear whether the material can switch multiple times, and fast enough to be useful for practical applications. From this perspective, while this work is a useful addition to literature, I do not believe it represents the type of novel advance suitable for Nature Communications. Besides concerns about novelty, I have the following major technical concerns:

1, The TDTR measures the cross-plane thermal conductivity, while the van der Pauw method measures the in-plane electrical conductivity, meaning the electric and thermal transport are measured in orthogonal directions. Will it make sense to estimate the electronic contribution to the total thermal conductivity without considering the anisotropy of either electric or thermal transport?

2, I notice a poor reversibility in many oxides whose thermal conductivity can be modulated by intercalation or deintercalation of hydrogen/oxygen ions via electrolyte gating, such as in SCO [ref 10] and WO_3 [Adv. Mater. 2019, 31, 1903738.]. I wonder how the thermal conductivity evolve over multiple cycles of gating in this LSCO system? In addition, what is the speed of switching? These are important aspects to demonstrate for a dynamic thermal switch. By the way, the author may want to add the WO_3 results into their Figure 3e.

3, A Pt layer, instead of Al, with thickness of 70 nm was used as the transducer layer in the TDTR analysis to avoid Al-gettering-induced oxygen deficiency in LSCO. I wonder if this thickness is large enough for Pt to effectively absorb the light? I know that thickness would be safe for Al, but am not sure whether it works for Pt. In addition, the authors sputtered the 70-nm Pt transducer layer after gating. How much will this process (vacuum condition, plasma, heating, etc) affect the oxygen stoichiometry of the gated sample?

Reviewer #2 (Remarks to the Author):

The authors present their results on the tunability of the thermal conductivity of $\text{La}_{0.5}\text{Sr}_{0.5}\text{CoO}_{3-\delta}$ films, by applying a gate voltage at room temperature. The authors report a 5-fold modulation of the thermal conductivity of the gated films, when compared to the ungated sample. The authors support their results with extensive characterization of the samples, including structural, electrical and thermal measurements. The structural and electrical measurements are in good agreement with previously reported results in the literature. The thermal conductivity results are being highlighted here, since they present a record modulation of the thermal conductivity of the gated samples. Their argument is supported by model analyses based on molecular dynamics and the Boltzmann transport

equation.

In general, the methodology followed by the authors is solid and the results that are presented seem robust with good, supported arguments and experiments. The key findings of this study are well supported by the TDTR measurements with the expected methodology followed and all the results and uncertainties are well within the acceptable limits of the technique.

The findings presented in the submitted manuscript are of substantial importance for applications within the scope of tunable thermal switches and thermal management in general, with broader applications. Nevertheless, there have been previously reported results on the same approach (10.1021/nl504505t and 10.1038/s41563-020-0612-0) which both have been cited by the authors in the manuscript.

The approach of the authors to the experiment is robust and they have followed the necessary methodology to prove their argument and supported with an extensive number of experiments and modeling. The main point of the manuscript – the modulation of the thermal conductivity of the gated samples is clearly stated and supported with good quality data, analyzing the underlying mechanisms of the results. The supplementary information is useful to further explain the experimental procedure and support their arguments and have been helpful in answering questions that were formed while reading the manuscript.

Comments:

- 1) The authors argue that the transition to the BM phase is non-volatile, and that the procedure is reversible. However, in line 341 of the main manuscript, when they discuss reverting the samples to the perovskite phase, the thermal conductivity is at 40% less than the as-deposited P-LSCO. This result hinders the argument of reversibility and their use in applications, as well as the argued “large tunability in the thermal conductivity of epitaxial LSCO films” (L401-402). Have the authors confirmed that the oxygen content is back to the desired levels to fully support the perovskite structure, either by means of electrical or x-ray diffraction measurements? The authors use the phrase “additional structural disorder” (L349); could the authors elaborate on that?
- 2) The temperature dependent measurements of the thermal conductivity of the films were performed in the range of 90 to 500K, under vacuum. Have the authors considered the possibility of creating additional oxygen vacancies at elevated temperatures during the measurements? The electrical resistivity in Fig. 1d reveals a difference between the gating of 3.0 V and 4.0 V throughout the whole range of the measurement temperature, however the results of the thermal conductivity between the two samples is similar according to Fig. S7.
- 3) The interfacial thermal conductance of the ungated Pt/LSCO (G1) and LSCO/LAO (G2) were determined to be 400 and 800 MW /m² K, respectively. The same result for the gated sample is G1 = 170 MW /m² K on average. To cross check the reasonableness of the measured G1 and G2, the authors have implemented the thickness-extrapolation method, but only for the ungated sample. Have the authors checked the results of G1 and G2 for the gated samples with the same method? Also, the authors claim that the low sensitivity at the G1, the determination on the thermal conductivity of the LSCO is not significantly impacted by G1. Could the authors give the maximum and minimum values of the thermal conductivity of the films for the enhanced uncertainty of G1?
- 4) The authors claim that the room temperature thermal conductivity modulation is “continuous” (L75, L240). However, according to the plot in Fig. 1a the values of the thermal conductivity seem to be separated to two groups of very similar values, with turning point the gate voltage of $V_g = 2V$, mostly driven by the topotactic reduction. For $V_g < 2V$, the values with the error bars are quite similar and for $V_g > 2V$, one can observe a saturated value for

the thermal conductivity with increasing voltage. Moreover, I find the phrase “RT modulation of the thermal conductivity of LSCO films by a remarkable factor of 5, a record value for such an approach” slightly misleading. It is true that the reduction in thermal conductivity is 5-fold according to the data, however, is not entirely reversible as stated in the manuscript, thus the term “reduction” instead of “modulation” would seem more appropriate. Moreover, Fig. 3e could indicate also the actual modulation of the thermal conductivity, in terms of reversibility.

The authors have provided a well-structured manuscript which clearly conveys the message of their experiments and their findings. The comparison with literature is very useful and helps transmitting the importance of the authors’ findings. The presented results are in good agreement with the literature and the references also help support some of their arguments, based on previous work from the authors and others.

Reviewer #3 (Remarks to the Author):

The authors investigated nanoscale films of LSCO and demonstrated continuous tuning of its thermal conductivity over a factor of 5 via changing the electric field. This large factor is made possible via two mechanisms - a room-temperature electrolyte-gate-induced non-volatile topotactic phase transformation and a metal-insulator transition. This result is significant and has reasonable implications for the thermal transport community.

1. Does the observed continuous tuning of k fit the results from an effective medium approximation?
2. It seems cycling will be a problem. How is the hysteresis over many cycles?
3. How is the thermal conductivity measured during gating? Does the electric field affect the properties of the transducer layer? Was the experiment performed with lasers passing through the ion-gating layer? If so, how is the model used to extract the k different from the usual TDTR measurement?
4. Can the authors comment on whether the heat flux calculation used in LAMMPS is suitable for this material? Did the authors verify its correctness and if the correct physics is captured?
5. The details for the BTE calculations seem to be missing. What are the potential and the unit cell used to account for the defects? Can the authors include the calculation details?

April 24, 2023

Responses to review comments on NCOMMS-22-42600

We are submitting our revised manuscript entitled “Wide-range continuous tuning of the thermal conductivity of $\text{La}_{0.5}\text{Sr}_{0.5}\text{CoO}_{3-\delta}$ films *via* room-temperature ion-gel gating”, after responding to, and making changes in light of, the comments from all three reviewers. We believe that we have adequately addressed all of the reviewers’ comments and concerns, and that the paper is yet further improved by the revisions. We appreciate the detailed and valuable comments provided by the reviewers. In what follows, we list the comments from the reviewer in italics and present our responses in regular font.

COMMENTS TO AUTHOR

Reviewer #1

In this paper, the authors report tunable thermal conductivity in epitaxial $\text{La}_{0.5}\text{Sr}_{0.5}\text{CoO}_{3-\delta}$ (LSCO) films under ion gel gating. Using a combination of XRD, electrical transport and TDTR measurements, they show that the topotactic phase transition from perovskite and brownmillerite phase, driven by the deintercalation of oxygen ions via electrolyte gating causes a decrease in cross-plane thermal conductivity, and claim to realize $\sim 5\times$ modulation in the conductivity.

The main idea demonstrated in this work is utilizing the migration of ions driven by electrolyte gating to dynamically control the thermal properties of oxide thin films, which, however, has been already reported in a couple of recent studies. Furthermore, the modulation by a factor of 5 seems questionable. As the authors mentioned in Line 347 on Page 17, after gating the BM-LSCO back to the P phase using -4.5 V, the thermal conductivity only increases to 2.7 from $0.85 \text{ W m}^{-1} \text{ K}^{-1}$,

meaning a tuning factor of ~3. Also, it is unclear whether the material can switch multiple times, and fast enough to be useful for practical applications. From this perspective, while this work is a useful addition to literature, I do not believe it represents the type of novel advance suitable for Nature Communications.

Response: We appreciate the concerns of Reviewer 1 regarding the modulation factor achieved, and the novelty of our work. Starting with the novelty issue, we certainly acknowledge (and in fact cite in our work) the couple of studies that have been done previously to control the thermal conductivity of oxides *via* electrolyte gating. However, it is worthwhile to summarize the key pieces of novelty that our study achieves over *all* prior work: the record tuning factor of >5 for a single-step bi-state process; the complete continuous tunability of this process; the fact that this is achieved at 300 K; and that the LSCO system allows this to be done with high air stability in both the BM *and* P states. These are very challenging with the SCO system emphasized in particular in past work, and the single-step tuning factor of >5 is also a record among all literature studies, regardless of the material types or tuning approaches.

Moving to the issue of the modulation factor, Reviewer 1 is certainly correct that reversibility, and endurance over many cycles, are challenges for the field. We believe this to be a general statement relevant to all work in the field. Frankly, thermal measurements are not the ideal first-pass characterization technique to use for probing and improving reversibility and endurance. High-throughput electrical transport measurements will surely lead the way here, along with structural techniques like XRD, which are also relatively easy and fast to perform. Some of the authors on this paper are currently heavily embroiled in such efforts, pursuing device design optimization, thin electrolytes, top gates, *etc.* We will eventually apply what we learn from such

studies to TDTR, but we consider this to be well beyond the scope of the current paper, which already establishes the record tuning of thermal conductivity across the first P to BM transformation. Indeed, we believe that full optimization of the reversibility and endurance will only evolve from a community effort, across many publications, with the ultimate limits being simply unknown right now. Nevertheless, in an effort to directly address the reviewer's concern, we have also improved the reversibility results we report in our revised manuscript. By better optimizing the process of gating back to the P phase at negative voltage, we have now gated from $P \rightarrow BM \rightarrow P$ with measured thermal conductivities at the three stages of $4.6 \pm 2.0 \text{ W m}^{-1} \text{ K}^{-1}$, $0.85 \pm 0.32 \text{ W m}^{-1} \text{ K}^{-1}$, and $3.5 \pm 1.1 \text{ W m}^{-1} \text{ K}^{-1}$, respectively. The latter value is improved from $2.7 \pm 0.5 \text{ W m}^{-1} \text{ K}^{-1}$, with certain optimization efforts (described below), and we believe it can be improved even further with better device design. We have thus already demonstrated 5.4-fold tuning of the thermal conductivity from P to BM and 4-fold tuning from BM to P, the largest tuning ratios achieved for a (reversible) bi-state transformation. We have provided more discussion on the reversibility in both the revised manuscript (see details in our response to Comment 2 below) and also the Supplementary Information (see details in our response to Comment 1 by Reviewer 2).

Regarding the final issue raised in this comment regarding the switching speed possible, we think the first key point to make is that the requirements on this for different applications vary greatly. Some applications have requirements as low as 1 Hz, while others require many kHz for competitiveness. For an electrochemical gating process such as the one studied here, there are two key contributions: the rate at which the electric double layer can be formed in the electrolyte, and the rate at which the P/BM phase transformation can then be driven through the oxide. For the first, the record for ion-gel electrolytes now lies in excess of a very impressive 1 MHz¹, which will surely therefore not be the limiting factor in our case. For the second factor (the $P \leftrightarrow BM$

transformation), the ultimate limits are not yet known, but certainly depend on the LSCO film thickness and device design. In the current study, we used typical thicknesses of ~45 nm (to enable reliable TDTR measurements of thin-film thermal conductivity), as well as side gates, yielding time scales of several minutes. We have already moved on to electrical work at film thickness of only 10 unit cells, however, as well as top gates, from which we expect substantially faster switching. We believe such work to be very interesting. However, again, this will require a community effort and is surely beyond the scope of the current paper.

Besides concerns about novelty, I have the following major technical concerns:

1. The TDTR measures the cross-plane thermal conductivity, while the van der Pauw method measures the in-plane electrical conductivity, meaning the electric and thermal transport are measured in orthogonal directions. Will it make sense to estimate the electronic contribution to the total thermal conductivity without considering the anisotropy of either electric or thermal transport?

Response: Due to the low measurement sensitivity to the in-plane thermal conductivity of LSCO thin films at these types of thicknesses², in addition to the challenges of probing the through-plane electrical resistivity, Reviewer 1 is correct that we were not able to measure the electronic and thermal transport along the same direction. However, considering that the thickness of our measured samples is ~45 nm, we can safely conclude that the electronic transport in our P-LSCO samples exhibits bulk behavior. Estimates of the electronic mean free path in this system yield nanometric scales. Also, from *ab initio* calculations of P-SrCoO₃ in a prior study³, the accumulated lattice thermal conductivity of P-SrCoO₃ is larger than 93% of its bulk value when the phonon

mean free path reaches 40 nm. In our P-LSCO system, with the additional alloying on the A site, the lattice thermal conductivity will approach the bulk limit at an even smaller film thickness. In addition, considering the cubic symmetry of P-LSCO, we anticipate isotropic band structures for both electrons and phonons. Thus, the through-plane thermal conductivity of P-LSCO, consisting of both electronic and lattice contributions, must be comparable to the in-plane thermal conductivity. We can further add that some of us have extensive experience of probing electronic transport properties of the P-LSCO system along different symmetry directions through in-plane measurements (*i.e.*, in-plane measurements of both (001) and (110) P-LSCO films), and any form of anisotropy is rarely seen. We are therefore very confident that our comparisons are valid.

As an additional point, since BM-LSCO is electrically insulating, thermal transport in BM-LSCO is mainly carried by phonons and will also approach the bulk limit. However, there is indeed possible anisotropy for thermal transport in BM-LSCO due to the orthorhombic symmetry of the BM phase. In this case, the large tuning factor of the LSCO thermal conductivity we demonstrated in this work is along the through-plane (*c*-axis) direction. To better clarify this point, we have modified our statement on Page 12 of the revised manuscript as:

“Coincident with this, the central experimental result of this work is that the through-plane thermal conductivity of LSCO decreases from 4.6 ± 2.0 to 0.85 ± 0.32 W m⁻¹ K⁻¹ as V_g varies from 0 to 4 V, *i.e.*, a factor of more than 5 reduction *via* ion-gel gating. We consider this record-high tuning factor to be for the through-plane thermal conductivity due to possible thermal transport anisotropy in the BM-LSCO film. However, it is reasonable to convert the in-plane electrical resistivity to the through-plane electronic thermal conductivity (see Supplementary Section 7 for more details).”

Also in the revised Supplementary Information, we have added a new Section 7 as follows:

“7. Comparison of in-plane vs. through-plane transport properties

Due to the low measurement sensitivity to the in-plane thermal conductivity of LSCO thin films at these types of thicknesses²⁶, in addition to the challenges of probing the through-plane electrical resistivity, we were unable to measure the electronic and thermal transport along the same direction. However, considering that the thickness of our measured samples is ~45 nm, we can safely conclude that the electronic transport in our P-LSCO samples exhibits bulk behavior. Estimates of the electronic mean free path in this system yield nanometric scales. Also, from *ab initio* calculations of P-SrCoO₃ in a prior study²⁷, the accumulated lattice thermal conductivity of P-SrCoO₃ is larger than 93% of its bulk value when the phonon mean free path reaches 40 nm. In our P-LSCO system, with the additional alloying on the A site, the lattice thermal conductivity will approach the bulk limit at an even smaller film thickness. In addition, considering the cubic symmetry of P-LSCO, we expect quite isotropic band structures for both electrons and phonons. Thus, the through-plane thermal conductivity of P-LSCO, consisting of both electronic and lattice contributions, must be comparable to the in-plane thermal conductivity.

Unlike P-LSCO, BM-LSCO is electrically insulating; therefore, thermal transport in BM-LSCO is mainly carried by phonons and will also approach the bulk limit. However, there is possible anisotropy for thermal transport in BM-LSCO due to the orthorhombic symmetry of the BM phase. In this case, the large tuning factor of the LSCO thermal conductivity we demonstrated in this work is along the through-plane (*c*-axis) direction.”

2. I notice a poor reversibility in many oxides whose thermal conductivity can be modulated by intercalation or deintercalation of hydrogen/oxygen ions via electrolyte gating, such as in SCO [ref 10] and WO₃ [Adv. Mater. 2019, 31, 1903738.]. I wonder how the thermal conductivity evolve

over multiple cycles of gating in this LSCO system? In addition, what is the speed of switching? These are important aspects to demonstrate for a dynamic thermal switch. By the way, the author may want to add the WO₃ results into their Figure 3e.

Response: We have comprehensively responded to these points above (see our first response text). We would add that we think it is important to treat O-based and H-based effects separately, given their different thermodynamics and kinetics. In that respect our results are a little simpler than some prior work (on SrCoO_x), as we have no evidence for any meaningful role for H; we can thus focus solely on O dynamics in this system. As noted above, we now have improved data on reversibility of the thermal conductivity modulation of LSCO in our revised manuscript. We have made changes accordingly on Page 17 of the revised manuscript to reflect this improvement:

“The thermal measurement of one such film generated a recovered P-LSCO thermal conductivity of $3.5 \pm 1.1 \text{ W m}^{-1} \text{ K}^{-1}$ at RT. This ~20% reduction in the Λ_{tot} of recovered P-LSCO compared to the as-deposited P-LSCO film no doubt arises from additional structural disorder induced during the cyclic gating (see Supplementary Section 8 for more discussion). Nevertheless, such data confirm the overall reversibility of the approach presented here. Further improvement of reversibility could be achieved through better device design and optimized gating conditions.”

Also, following the reviewer’s suggestion, we have updated Fig. 3e to include the WO₃ results from [Adv. Mater. 2019, 31, 1903738] in the revised manuscript (cited as Ref. [28]), as shown below.

Fig. 3 | Impact of V_g on thermal properties. **a** The thermal conductivity (left axis) and electrical resistivity (right axis) of LSCO films, **b** the interfacial thermal conductance between Pt and LSCO, and **c** oxygen non-stoichiometry due to vacancies, as functions of V_g . In panel **a**, Λ_{tot} represents the measured total thermal conductivity of the LSCO films. Λ_{el} represents the electronic thermal conductivity estimated from electrical conductivity based on the Wiedemann-Franz law. The black dash-dotted lines represent the upper and lower limits of Λ_{el} based on different Lorenz numbers (L). The dashed lines serve as guides to the eye. In panel **c**, $\delta = 0.5$ (blue circles) is assumed for the complete transition to brownmillerite ($V_g \geq 3$ V), as our method used to estimate δ (red squares) is not valid above $\delta \approx 0.25$ (see Methods). The red-blue graded dashed line serves as a guide to the eye. **d** MD simulation results for the lattice thermal conductivity of LSCO with different δ . In all panels, the vertical pink and blue dashed lines represent, respectively, the starting and ending points of the P \rightarrow P + BM \rightarrow BM transformations. **e** Comparison of thermal conductivity tuning factor through bi-state tuning process in this work and previous experimental studies^{10,19-28}, including $PbZr_{0.3}Ti_{0.7}O_3$ (PZTO, 5 cycles)^{20,21}, $PbZrO_3$ (PZO, 10+ cycles)²², $PbTiO_3$ (PTO)²³, $BiFeO_3$ (BFO)²⁴, bio-polymers (B-P, 10+ cycles)²⁵, azobenzene polymers (A-P, 6 cycles)²⁶, liquid crystal networks (LCN)²⁷, Li_xCoO_2 (LCO, 2 cycles)¹⁹, WO_3 (WO, 3 cycles)²⁸, and $SrCoO_{3-\delta}$ (SCO, 1 cycle, and H-SCO)¹⁰. Here, one cycle is defined as the transformation from state/phase A to state/phase B, and then return to state/phase A. Cycle numbers are provided here when available from the literature.

3. A Pt layer, instead of Al, with thickness of 70 nm was used as the transducer layer in the TDTR analysis to avoid Al-gettering-induced oxygen deficiency in LSCO. I wonder if this thickness is

large enough for Pt to effectively absorb the light? I know that thickness would be safe for Al, but am not sure whether it works for Pt. In addition, the authors sputtered the 70-nm Pt transducer layer after gating. How much will this process (vacuum condition, plasma, heating, etc) affect the oxygen stoichiometry of the gated sample?

Response: In response to the first point, at the laser wavelength used in our TDTR measurements (783 nm), the extinction coefficients of Pt and Al are 7.88 and 8.45, respectively^{4,5}. This corresponds to an optical penetration depth of 7.9 nm in Pt and 7.4 nm in Al. Therefore, with a thickness of 70 nm, both Al and Pt transducers are optically opaque and can effectively absorb the light at 783 nm. Regarding the second point raised, generally, Al is the most commonly used transducer for TDTR measurements. However, it is now acknowledged in the complex oxide thin film community that Al is an effective O scavenger. Deposition of Al, even at room temperature, can lead to some significant reduction of the thickness into underlying perovskite films⁶. At least the thermodynamic driving force for this is illustrated by the enthalpy of formation of Al₂O₃ ($-1675.7 \text{ kJ mol}^{-1}$)⁷, which can be compared to that of PtO₂ (around -80 kJ mol^{-1})⁸, indicating that Al has a much larger driving force to getter oxygen. We therefore chose Pt over Al as our transducer, to be safe. Regarding the sputtering process for Pt, we have verified many times that conditions such as those used in that process have negligible impact on the LSCO to any significant fraction of its 45 nm thickness. Reduction of LSCO films in vacuum requires significant temperature increase.

To better clarify the Pt transducer properties, we have included the following sentence in the revised manuscript (Page 23):

“At the laser wavelength of 783 nm, the extinction coefficient is ~ 7.4 for Pt⁶⁹; therefore, the thickness of 70 nm is sufficient for the Pt transducer to be considered optically opaque.”

Reviewer #2

The authors present their results on the tunability of the thermal conductivity of $\text{La}_{0.5}\text{Sr}_{0.5}\text{CoO}_{3-\delta}$ films, by applying a gate voltage at room temperature. The authors report a 5-fold modulation of the thermal conductivity of the gated films, when compared to the ungated sample. The authors support their results with extensive characterization of the samples, including structural, electrical and thermal measurements. The structural and electrical measurements are in good agreement with previously reported results in the literature. The thermal conductivity results are being highlighted here, since they present a record modulation of the thermal conductivity of the gated samples. Their argument is supported by model analyses based on molecular dynamics and the Boltzmann transport equation.

In general, the methodology followed by the authors is solid and the results that are presented seem robust with good, supported arguments and experiments. The key findings of this study are well supported by the TDTR measurements with the expected methodology followed and all the results and uncertainties are well within the acceptable limits of the technique.

The findings presented in the submitted manuscript are of substantial importance for applications within the scope of tunable thermal switches and thermal management in general, with broader applications. Nevertheless, there have been previously reported results on the same approach

(10.1021/nl504505t and 10.1038/s41563-020-0612-0) which both have been cited by the authors in the manuscript.

The approach of the authors to the experiment is robust and they have followed the necessary methodology to prove their argument and supported with an extensive number of experiments and modeling. The main point of the manuscript – the modulation of the thermal conductivity of the gated samples is clearly stated and supported with good quality data, analyzing the underlying mechanisms of the results. The supplementary information is useful to further explain the experimental procedure and support their arguments and have been helpful in answering questions that were formed while reading the manuscript.

Response: We greatly appreciate these positive and supportive comments from Reviewer 2.

1) The authors argue that the transition to the BM phase is non-volatile, and that the procedure is reversible. However, in line 341 of the main manuscript, when they discuss reverting the samples to the perovskite phase, the thermal conductivity is at 40% less than the as-deposited P-LSCO. This result hinders the argument of reversibility and their use in applications, as well as the argued “large tunability in the thermal conductivity of epitaxial LSCO films” (L401-402). Have the authors confirmed that the oxygen content is back to the desired levels to fully support the perovskite structure, either by means of electrical or x-ray diffraction measurements? The authors use the phrase “additional structural disorder” (L349); could the authors elaborate on that?

Response: The first comment here is essentially identical to the overall comment of Reviewer 1. As noted in response to that comment (please see above, Pages 2-4), our efforts to improve the gating back to the P phase after reduction to BM have already generated a ~40% improvement in the recovered thermal conductivity of the P phase (from 2.7 ± 0.5 to 3.5 ± 1.1 W m⁻¹ K⁻¹). We now report modulation of the thermal conductivity over a factor of 5.4 from P to BM, and over a factor of 4.1 from BM to P (improved from a factor of 3.2). As also remarked above, we believe there is much scope for further improvement of this, *via* better device design, *etc.*

With respect to the O content of the recovered P phase, and the “additional structural disorder”, please consider Fig. R1 below. This shows (Fig. R1a,b) high-resolution XRD measurements on an ungated sample, after +3 V, our previous -4.5 V case, and our new and improved -4.5 V case. The latter derives simply from increased gating time at the largest negative voltages. The ideal situation after negative V_g would be a P(002) reflection exactly overlapping with the ungated peak. In reality, we mostly return to the ungated P(002) location from the BM(008) position, but with a clear remnant shift to the left in Fig. R1b (*i.e.*, a larger P lattice parameter). Explicitly, the BM(006) and (0010) peaks are entirely extinguished, definitively establishing that long-range V_O order is annihilated, but the lattice parameter of the recovered P phase remains expanded. This means that the V_O concentration is higher than it was originally, although the “new” reverse-gated sample is clearly better in this respect than the “original” reverse-gated sample. This is part of what we mean by additional disorder, and it can be further quantified through electronic transport (Fig. R1c). Specifically, the approach described in the main text and Methods of our manuscript to determine the O non-stoichiometry δ from transport yields 0.11 for the as-deposited P-LSCO film, 0.18 for the “original” reverse-gated film (following P \rightarrow BM \rightarrow P), and 0.16 for the “new” reverse-gated film. We believe this is the primary reason for the improved thermal

conductivity of the “new” cycled P-LSCO film. We would add that in synchrotron XRD we have performed, it is clear that the overall epitaxial quality of the as-deposited P-LSCO films exceeds those of the P-LSCO films recovered after gating to BM, as judged by peak widths and Laue fringes. This is unsurprising after topotactic transformation from P to BM back to P, and no doubt also plays some role in the reduced thermal conductivity.

Fig. R1 | XRD measurements on a reverse-gated sample. a Wide-angular-range XRD results from LSCO devices on LAO(001) substrates after gating to +3.0 V and after reverse gating to -4.5 V (both the original and new samples). The y-axis is on a logarithmic scale. **b** An enlarged version of the same scans as in a, around the film and substrate perovskite (002) peak (BM (008)), with the intensity axis (log scale) normalized to the substrate intensity, to better show the changes in the film position with reverse (*i.e.*, negative- V_g) gating. **c** Temperature-dependent electrical resistivity measurements of the as-deposited P-LSCO, after gating to +3.0 V, and after reverse gating to -4.5 V (the color code is the same for all panels).

To better address the reversibility issue, we have added the above XRD and electrical characterizations for the new reverse-gated LSCO sample in a new Section 8 of the revised Supplementary Information:

“8. Reversibility of tuning the LSCO thermal conductivity

To explore the reversible nature of the V_g -induced $P \leftrightarrow BM$ phase transformation in LSCO, we also reverted some gated BM-LSCO films to the P phase by applying reverse gate voltages of up to -4.5 V at RT. The RT thermal conductivity of one reverse-gated P-LSCO film ($P \rightarrow BM \rightarrow P$) was measured to be 3.5 ± 1.1 W m⁻¹ K⁻¹. This $\sim 20\%$ reduction in the Λ_{tot} of recovered P-LSCO compared to the as-deposited P-LSCO film arises from additional structural disorder induced during the cyclic gating. This can be seen from the high-resolution XRD measurements (Fig. S8a,b) on the as-deposited P-LSCO film (labeled as “ungated”), after gating to $+3.0$ V (*i.e.*, transformed to BM), and after reverse gating to -4.5 V (*i.e.*, recovered to P). The ideal situation after negative V_g would be a P(002) reflection exactly overlapping with the ungated peak. In reality, the return to the ungated P(002) location from the BM(008) position has a remnant shift to the left in Fig. S8b (*i.e.*, a larger P lattice parameter). Explicitly, the BM(006) and (0010) peaks are entirely extinguished, definitively establishing that long-range V_O order is annihilated, but the lattice parameter of the recovered P phase remains expanded. This indicates that the V_O concentration is higher than it was originally for the ungated sample. This additional structural disorder can be further quantified through electronic transport (Fig. S8c). Specifically, the approach described in the main text and Methods of our paper to determine the O non-stoichiometry δ from transport yields 0.11 for the as-deposited P-LSCO film and 0.16 for the reverse-gated film (following $P \rightarrow BM \rightarrow P$). This additional disorder is unsurprising after topotactic transformation from P to BM back to P, and no doubt plays some role in the reduced thermal conductivity. Nevertheless, such data confirm the overall reversibility of the approach presented here. Further improvement of reversibility can be achieved through better device design and optimized gating conditions.

Fig. S8 | XRD measurements on a reverse-gated sample. a Wide-angular-range XRD results from LSCO devices on LAO(001) substrates after gating to +3.0 V and after reverse gating to -4.5 V. The y-axis is on a logarithmic scale. **b** An enlarged version of the same scans as in a, around the film and substrate perovskite (002) peak (BM (008)), with the intensity axis (log scale) normalized to the substrate intensity, to better show the changes in the film position and intensity with reverse gating. **c** Temperature-dependent electrical resistivity measurements of the as-deposited P-LSCO, after gating to +3.0 V, and after reverse gating to -4.5 V (the color code is the same for all panels).”

2) The temperature dependent measurements of the thermal conductivity of the films were performed in the range of 90 to 500K, under vacuum. Have the authors considered the possibility of creating additional oxygen vacancies at elevated temperatures during the measurements? The electrical resistivity in Fig. 1d reveals a difference between the gating of 3.0 V and 4.0 V throughout the whole range of the measurement temperature, however the results of the thermal conductivity between the two samples is similar according to Fig. S7.

Response: For thermal measurements at elevated temperatures (from RT to ~500 K), all experiments were conducted in air. Further introduction of oxygen vacancies into LSCO is not likely in such an oxygen-containing environment at these temperatures; higher temperatures would be needed. We also anticipate the Pt transducer to serve as a protection layer preventing additional reduction. Moreover, and this is truly the “bottom line” on this issue, we performed careful checks by heating ungated LSCO samples to 500 K, cooling to RT, and remeasuring the thermal conductivity, which turned out to be identical to the ungated sample prior to heating. Therefore, we confirmed directly that no significant density of additional oxygen vacancies was introduced at elevated temperatures during our thermal measurements.

With respect to the second point raised, due to the significantly higher electrical resistivity of BM-LSCO, the converted electronic thermal conductivities are nearly zero for the samples gated to both 3 and 4 V, over the entire temperature range, despite their different electrical resistivities. This suggests that the contribution from electrons to thermal transport is negligible, and phonons are the primary heat carriers for BM-LSCO. With the same crystallographic phase and similar structural features, we thus expect the thermal conductivities to be close for the two samples gated to 3 and 4 V, as illustrated in Fig. S7.

To improve the clarity on the measurement details, we modified the statement on Page 24 in the Methods as:

“For temperature-dependent TDTR measurements over the range of ~90 to ~500 K, the sample was mounted on a heating/cooling stage. Specifically, measurements at low temperatures (from ~90 K to RT) were done under vacuum, while measurements at elevated temperatures (from RT to ~500 K) were carried out in air.”

3) The interfacial thermal conductance of the ungated Pt/LSCO (G_1) and LSCO/LAO (G_2) were determined to be 400 and 800 MW/m² K, respectively. The same result for the gated sample is $G_1 = 170$ MW/m² K on average. To cross check the reasonableness of the measured G_1 and G_2 , the authors have implemented the thickness-extrapolation method, but only for the ungated sample. Have the authors checked the results of G_1 and G_2 for the gated samples with the same method? Also, the authors claim that the low sensitivity at the G_1 , the determination on the thermal conductivity of the LSCO is not significantly impacted by G_1 . Could the authors give the maximum and minimum values of the thermal conductivity of the films for the enhanced uncertainty of G_1 ?

Response: We thank Reviewer 2 for raising this concern about the thermal interfacial conductances of the gated samples. Since G_1 and Λ_{LSCO} both decrease as the gate voltage increases, the measurement sensitivities to G_1 and Λ_{LSCO} of the ungated samples are smaller than those of gated samples. In addition, due to the larger Λ_{LSCO} of the ungated samples, the measurement sensitivity to G_2 is larger for the ungated samples, due to the increase in the thermal penetration depth ($h = \sqrt{\Lambda/(\pi C f)}$, where C is the heat capacity and f is the modulation frequency). Therefore, we conducted dual-frequency TDTR measurements with improved measurement sensitivity to G_2 to extract reasonable values of G_2 for the ungated samples (see Supplementary Fig. S3). To address the issue of low measurement sensitivity to G_1 , we relied on the thickness-extrapolation approach to cross check G_1 and G_2 for the ungated samples. While for the gated samples, with the decrease in G_1 and Λ_{LSCO} , the measurement sensitivities to both parameters are significantly enhanced, allowing for the reliable determination of G_1 and Λ_{LSCO} via simultaneous fitting of routine TDTR measurements. Therefore, there is no great need for us to apply the thickness-extrapolation approach for the gated samples.

Since the ungated LSCO films are the most challenging samples for thermal measurements due to their largest Λ_{LSCO} among all samples, we take the ungated LSCO as an example to justify our statement that the impact of G_1 on Λ_{LSCO} is negligible for the ungated samples. Table R1 lists the values of G_1 and Λ_{LSCO} obtained from simultaneous fitting, in comparison with those obtained from fitting for Λ_{LSCO} only with G_1 fixed as $400 \text{ MW m}^{-2} \text{ K}^{-1}$. As shown in Table R1, due to the low measurement sensitivity to G_1 , the fitted G_1 varies from 250 to $920 \text{ MW m}^{-2} \text{ K}^{-1}$, while the fitted Λ_{LSCO} only varies from 4.1 to $5.5 \text{ W m}^{-1} \text{ K}^{-1}$. If we fix G_1 at $400 \text{ MW m}^{-2} \text{ K}^{-1}$, the obtained Λ_{LSCO} (listed in the last column of Table R1) varies from 4.1 to $6.3 \text{ W m}^{-2} \text{ K}^{-1}$. The variations of Λ_{LSCO} with or without G_1 being fixed in the data analysis are all within the uncertainty. Thus, it clearly shows that the fitting values of Λ_{LSCO} are not impacted by G_1 for the ungated samples.

Table R1. Results of G_1 and Λ_{LSCO} for the ungated samples measured at different locations and modulation frequencies, obtained from simultaneous fitting for both G_1 and Λ_{LSCO} or fitting for Λ_{LSCO} only ($G_1 = 400 \text{ MW m}^{-2} \text{ K}^{-1}$).

Sample #	Location #	Modulation frequency (MHz)	G_1 ($\text{MW m}^{-2} \text{ K}^{-1}$)	Λ_{LSCO} ($\text{W m}^{-1} \text{ K}^{-1}$)	G_1 ($\text{MW m}^{-2} \text{ K}^{-1}$)	Λ_{LSCO} ($\text{W m}^{-1} \text{ K}^{-1}$)
1	L1	1.5	920	4.7	400	6.3
		9	490	4.3		4.3
		18.8	240	5.0		5.1
	L2	1.5	280	5.5		4.5
		9	530	4.2		4.1
		18.8	320	4.6		4.7
	L3	1.5	390	4.6		4.5
		9	600	4.5		4.3
		18.8	410	4.1		4.1
2	L1	1.5	520	4.3	4.9	
		9	660	4.1	4.3	
		18	330	4.9	5.0	
	L2	1.5	440	4.7	4.9	
		9	250	4.9	4.4	
		18	440	4.5	4.5	
Average (range)			455 (240 - 920)	4.6 (4.1 - 5.5)	400 (fixed)	4.7 (4.1 - 6.3)

To better demonstrate that the determination of Λ_{LSCO} is not significantly impacted by G_1 , we have added Table R1 as Table S2 in the revised Supplementary Information (Pages 8-9), together with the following discussion:

“Due to the low sensitivity to G_1 , the determination of Λ_{LSCO} is not significantly impacted by G_1 . Table S2 summarizes the values of G_1 and Λ_{LSCO} from measurement fitting. When both parameters are fitted simultaneously, G_1 varies from 250 to 920 MW m⁻² K⁻¹, while Λ_{LSCO} only varies from 4.1 to 5.5 W m⁻¹ K⁻¹. When G_1 is fixed at 400 MW m⁻² K⁻¹, the fitted Λ_{LSCO} (listed in the last column of Table S2) varies from 4.1 to 6.3 W m⁻² K⁻¹. The variations of Λ_{LSCO} with or without G_1 being fixed in the data analysis are all within the measurement uncertainty.”

4) The authors claim that the room temperature thermal conductivity modulation is “continuous” (L75, L240). However, according to the plot in Fig. 3a the values of the thermal conductivity seem to be separated to two groups of very similar values, with turning point the gate voltage of $V_g = 2\text{V}$, mostly driven by the topotactic reduction. For $V_g < 2\text{V}$, the values with the error bars are quite similar and for $V_g > 2\text{V}$, one can observe a saturated value for the thermal conductivity with increasing voltage. Moreover, I find the phrase “RT modulation of the thermal conductivity of LSCO films by a remarkable factor of 5, a record value for such an approach” slightly misleading. It is true that the reduction in thermal conductivity is 5-fold according to the data, however, is not entirely reversible as stated in the manuscript, thus the term “reduction” instead of “modulation” would seem more appropriate. Moreover, Fig. 3e could also indicate the actual modulation of the thermal conductivity, in terms of reversibility.

Response: We respond first to the point about the reversibility of the thermal conductivity modulation. This is the same point we responded to above twice (Reviewer 1, overall comments and Comment 2, and Reviewer 2, Comment 1). We do not repeat our arguments here, including that we now improve on the reversibility level in our revised manuscript. In response to this point, however, we have replaced “modulation” with “tuning” in the revised manuscript. Also following Reviewer 2’s suggestion, we now explicitly show the reversibility of the various tuning factors in the caption to Fig. 3e, in all cases where multiple cycle data were reported (including ours).

Regarding the continuous nature of the modulation, we first note that part of the issue here may simply be the non-negligible error bars in Figure 3a. Beyond this, we note that, as pointed out in the manuscript, the P to BM transformation is a first-order one, and should thus exhibit some discontinuity, as well as the P+BM phase coexistence that is prominent here. Due to the latter, we still expect continuous tunability of the thermal conductivity, however, as the phase fractions of P and BM change across the transition. In fact, even the structure of P-LSCO continuously changes with V_g in the mixed phase region (as δ is not constant). Unfortunately, it is unavoidable that the measurement error bars on Λ_{LSCO} are relatively large when V_g is low, resulting from the significantly suppressed measurement sensitivity to Λ_{LSCO} .

The authors have provided a well-structured manuscript which clearly conveys the message of their experiments and their findings. The comparison with literature is very useful and helps transmitting the importance of the authors’ findings. The presented results are in good agreement with the literature and the references also help support some of their arguments, based on previous work from the authors and others.

Response: Again, we thank Reviewer 2 for their supportive comments on our work.

Reviewer #3

The authors investigated nanoscale films of LSCO and demonstrated continuous tuning of its thermal conductivity over a factor of 5 via changing the electric field. This large factor is made possible via two mechanisms - a room-temperature electrolyte-gate-induced non-volatile topotactic phase transformation and a metal-insulator transition. This result is significant and has reasonable implications for the thermal transport community.

1. Does the observed continuous tuning of k fit the results from an effective medium approximation?

Response: We appreciate the suggestion of fitting thermal conductivity results with an effective medium approximation (EMA) model. However, the P to BM transformation is substantially more complex and subtle than a typical A to B transition. In particular, it is not just the volume fractions of P and BM phases that vary across the transition, but also the stoichiometry of the phases. Specifically, the P-LSCO phase continuously decreases in oxygen content as the gate voltage is increased in the mixed phase region; therefore, it is challenging (if not completely impossible) to pin down the oxygen stoichiometry of the P-LSCO and the percentages of the P and BM phases. Even if we had such information, the relationship is still nontrivial between the thermal conductivity of P-LSCO and its local oxygen stoichiometry at any point during gating. This is because the EMA model only considers “mass loss” at a large scale, while in reality LSCO experiences additional “bonding loss” and phonon scattering with oxygen vacancies as well. In addition, due to the coexistence of P and BM phases in the transition regime, boundaries necessarily form between the phases, and scatter heat-carrying phonons, which cannot be captured by the EMA. Even if we apply the EMA only to the as-deposited P-LSCO and BM-LSCO, by

assuming all oxygen vacancies as “pores”, an increase of ~10% in the “porosity” from P ($\delta = 0.1$) to BM ($\delta = 0.5$) only leads to a reduction of ~15% in the thermal conductivity, which cannot explain our experimental observation. Thus, we choose not to apply the EMA to fit the thermal conductivity data of our samples.

2. It seems cycling will be a problem. How is the hysteresis over many cycles?

Response: We agree with Review 3 that cycling (reversibility and endurance) will be a key challenge for applications of the thermal conductivity modulation of oxides (including LSCO) *via* electrolyte gating and related approaches. Similar concerns were also raised by Reviewers 1 and 2. Please refer to our responses to the overall comments and Comment 2 from Reviewer 1, and Comment 1 from Reviewer 2.

3. How is the thermal conductivity measured during gating? Does the electric field affect the properties of the transducer layer? Was the experiment performed with lasers passing through the ion-gating layer? If so, how is the model used to extract the k different from the usual TDTR measurement?

Response: One of the great advantages of the non-volatile nature for electrolyte gating of LSCO is that it enables us to conduct thermal measurements using an *ex-situ* configuration, which mitigates the difficulties of measuring the LSCO thermal conductivity during gating. In our work, a set of as-deposited P-LSCO samples was prepared and gated, separately, to different gate voltages (up to 4 V). After the gating was completed, the information of phase composition and

stoichiometry was locked within the samples (due to the non-volatile nature) and the ion gel was removed. Then we sputtered Pt on top of all the samples as the transducer and proceeded with TDTR measurements. To improve clarity around this issue, we have modified the statement on Page 10 of the manuscript as:

“The thermal conductivities of LSCO thin films gated to various V_g (from 0 to 4 V) were measured with the TDTR approach, which is an ultrafast-laser-based pump-probe technique^{38,39}. All TDTR measurements were made *ex-situ*, after biasing and ion-gel removal, taking advantage of the non-volatile nature of the electrolyte gating in this case^{10-12,14,29}.”

4. Can the authors comment on whether the heat flux calculation used in LAMMPS is suitable for this material? Did the authors verify its correctness and if the correct physics is captured?

Response: We thank the reviewer for raising this interesting and important question. As pointed out by Fan, *et al.*⁹, the heat flux calculation in LAMMPS could be problematic if the potential is in a many-body form. The potentials used in LAMMPS in this work are the short-range Buckingham potential + long-range Coulombic potential, which are two-body potentials. Therefore, we expect that the heat flux formalism in LAMMPS should be reliable. The details of the potential usage description for this material are fully presented in our prior work¹⁰ (Section 4 and Supporting Information Section S4).

5. The details for the BTE calculations seem to be missing. What are the potential and the unit cell used to account for the defects? Can the authors include the calculation details?

Response: We appreciate the reviewer's question. Similar to the LAMMPS-based MD calculations, the details of our BTE calculations are also exactly the same as those presented in our previous work¹⁰. The phonon frequency and group velocity are calculated using a Lattice Dynamics program with a hybrid interatomic potential, which is described by the core-shell model, short-range Buckingham potential, and long-range Coulombic forces developed by Read, *et al.*¹¹. The acoustic branches agree well with the first principles and experimental data. The defects are included as perturbations to the pure crystal for phonon-defect scattering only. We do not include any defects for phonon dispersion as it is too computationally expensive. Considering that the classical potential is not as accurate as first-principles calculations, nor can they correctly predict optical branches, we acknowledge that our procedures might not be ideal. However, our goal here is to provide qualitative insights to better interpret the experiment results, instead of pursuing super-high accuracy in the BTE defect calculations. For all the details of the BTE calculations, please refer to Section 4 and Supporting Information Section S3 of Ref. [10].

References

- 1 Zare Bidoky, F. & Frisbie, C. D. Sub-3 V, MHz-class electrolyte-gated transistors and inverters. *ACS Appl. Mater. Interfaces* **14**, 21295-21300 (2022).
- 2 Zhu, J. *et al.* Revealing the origins of 3D anisotropic thermal conductivities of black phosphorus. *Adv. Electron. Mater.* **2**, 1600040 (2016).
- 3 Lu, Q. *et al.* Bi-directional tuning of thermal transport in SrCoO_x with electrochemically induced phase transitions. *Nat. Mater.* **19**, 655-662 (2020).
- 4 Werner, W. S., Glantschnig, K. & Ambrosch-Draxl, C. Optical constants and inelastic electron-scattering data for 17 elemental metals. *J. Phys. Chem. Ref. Data* **38**, 1013-1092 (2009).
- 5 Rakić, A. D. Algorithm for the determination of intrinsic optical constants of metal films: application to aluminum. *Appl. Optics* **34**, 4755-4767 (1995).
- 6 Homonnay, N. *et al.* Interface reactions in LSMO-metal hybrid structures. *ACS Appl. Mater. Interfaces* **7**, 22196-22202 (2015).
- 7 National Institute of Standards and Technology, *NIST Chemistry WebBook, SRD 69*, <https://webbook.nist.gov/cgi/cbook.cgi?ID=C1344281&Mask=2> (2021).
- 8 Nagano, Y. Standard enthalpy of formation of platinum hydrous oxide. *J. Therm. Anal. Calorim.* **69**, 831-839 (2002).
- 9 Fan, Z. *et al.* Force and heat current formulas for many-body potentials in molecular dynamics simulations with applications to thermal conductivity calculations. *Phys. Rev. B* **92**, 094301 (2015).
- 10 Wu, X. *et al.* Glass-like through-plane thermal conductivity induced by oxygen vacancies in nanoscale epitaxial La_{0.5}Sr_{0.5}CoO_{3-δ}. *Adv. Funct. Mater.* **27**, 1704233 (2017).
- 11 Read, M. S., Islam, M. S., Watson, G. W., King, F. & Hancock, F. E. Defect chemistry and surface properties of LaCoO₃. *J. Mater. Chem.* **10**, 2298-2305 (2000).

REVIEWER COMMENTS

Reviewer #1 (Remarks to the Author):

I have gone through the response that the authors made to my and the other reviewers' comments. I think the revised manuscript has been largely improved. I would like to recommend it for publication.

Reviewer #2 (Remarks to the Author):

The authors have provided sufficient explanations and changes to the original manuscript to address my questions and doubts during the first round of reviewing the manuscript. I believe that, overall, this is a well-rounded study that will help the advancement of collective knowledge in the field of thermal transport of thin films, and how one can take advantage of the P to BM transition for, eventual/future, applications.

Reviewer #3 (Remarks to the Author):

The authors have clarified most of my questions.

I have looked at the supplementary information cited in their response. I find them useful and should be summarized in the SI for this manuscript. The assumptions that are implicit in these calculations should be stated. For example, they should add statements regarding a) "classical potential is not as accurate as first-principles calculations, nor can they correctly predict optical branches, we acknowledge that our procedures might not be ideal. However, our goal here is to provide qualitative insights to better interpret the experiment results, instead of pursuing super-high accuracy in the BTE defect calculations." b) "The potentials used in LAMMPS in this work are the short-range Buckingham potential + long-range Coulombic potential, which are two-body potentials. Therefore, we expect that the heat flux formalism in LAMMPS should be reliable." If the authors can add NEMD simulations to support their EMD results, they can make their MD results more solid.

For the sensitivity of the interfacial thermal conductance to the measurement, it will be clearer if the sensitivity curve is presented. Are there any known methods to verify the correctness of such a multivariate fit ?

Overall, I think the results are sound and interesting for the scientific community.

Responses to review comments on NCOMMS-22-42600A

We are resubmitting our revised manuscript entitled “Wide-range continuous tuning of the thermal conductivity of $\text{La}_{0.5}\text{Sr}_{0.5}\text{CoO}_{3-\delta}$ films *via* room-temperature ion-gel gating”, after making a second round of revisions, in response to the comments from all three reviewers. We believe that we have adequately addressed the technical comments and concerns from Reviewer 3. In what follows, we list the reviewers’ comments in italics and present our responses in regular font.

Reviewer #1 (Remarks to the Author):

I have gone through the response that the authors made to my and the other reviewers' comments. I think the revised manuscript has been largely improved. I would like to recommend it for publication.

Reviewer #2 (Remarks to the Author)

The authors have provided sufficient explanations and changes to the original manuscript to address my questions and doubts during the first round of reviewing the manuscript. I believe that, overall, this is a well-rounded study that will help the advancement of collective knowledge in the field of thermal transport of thin films, and how one can take advantage of the P to BM transition for, eventual/future, applications.

Response: We are delighted that Reviewers 1 and 2 are satisfied with our first round of revisions, and we appreciate these positive and supportive comments from both reviewers.

Reviewer #3 (Remarks to the Author):

The authors have clarified most of my questions.

I have looked at the supplementary information cited in their response. I find them useful and should be summarized in the SI for this manuscript. The assumptions that are implicit in these calculations should be stated. For example, they should add statements regarding a) "classical potential is not as accurate as first-principles calculations, nor can they correctly predict optical branches, we acknowledge that our procedures might not be ideal. However, our goal here is to provide qualitative insights to better interpret the experiment results, instead of pursuing super-high accuracy in the BTE defect calculations." b) "The potentials used in LAMMPS in this work are the short-range Buckingham potential + long-range Coulombic potential, which are two-body potentials. Therefore, we expect that the heat flux formalism in LAMMPS should be reliable." If the authors can add NEMD simulations to support their EMD results, they can make their MD results more solid.

Response: We are thankful for Reviewer 3's constructive comments. Following the suggestions, we have added more discussions in the Supplementary Section 5:

"The potentials used in LAMMPS in this work are the short-range Buckingham potential + long-range Coulombic potential, which are two-body potentials. Therefore, we expect that the heat flux formalism in LAMMPS should be reliable. Classical potentials are not as accurate as first-principles calculations, nor can they correctly predict optical branches. However, our goal here is to provide qualitative insights to better interpret the experiment results, rather than pursuing very high accuracy in BTE calculations."

Also, we have performed NEMD simulations for some case studies to validate the equilibrium Green-Kubo MD (GKMD) results. For this purpose, we have selected four cases with $\delta = 0.1, 0.25, 0.375,$ and 0.5 that were studied by GKMD. The same potential settings are used in the NEMD calculation to keep things consistent. The thermal conductivity values obtained from NEMD agree reasonably well with the GKMD results, supporting the validity of our equilibrium MD simulation results. Accordingly, we have added some discussion about the NEMD simulations and the comparison between the GKMD and NEMD in the Supplementary Section 5 of the revised Supplementary Information:

“To validate the equilibrium GKMD results, we further performed non-equilibrium molecular dynamics (NEMD) for four cases with the non-stoichiometry δ values that have been studied with GKMD (*i.e.*, $\delta = 0.1, 0.25, 0.375,$ and 0.5). The NEMD simulation domain is shown in Fig. S6a with a size of $2.2938 \times 3.0584 \times 3.0584$ nm³. NEMD simulations were performed using the LAMMPS package with the same classical potential settings as implemented in the GKMD. The hot reservoir was set in the middle of the simulation domain with a length of 2 nm, and the cold reservoir of the same size was split into two pieces equally and placed at the two edges of the box along the x direction. In all the three dimensions, periodic boundary conditions were applied. A Langevin thermostat was adopted to maintain the temperatures of the hot reservoir at 598.15 K and the cold reservoir at 548.15 K. The timestep was set as 0.5 fs. In the simulation, the NVT ensemble was fixed first for 100,000 steps and then followed by an NVE ensemble for 100,000 steps to fully relax the lattice. Then the system was fixed at the NVE ensemble for another 200,000 steps, during which the temperature gradient and heat flow rate were recorded. Then the thermal conductivity can be obtained based on Fourier’s law. Figure S6b shows an example of the temperature gradient along the x -axis for the $\delta = 0.25$ case. It is noticed that the hot reservoir has

the highest temperature, while the cold reservoir exhibits the lowest temperature. The length between the hot and cold reservoir is 9.2 nm, different from that in Fig. S6a. This discrepancy is caused by the position where the temperature is monitored not being precisely at the boundary of the reservoirs. Figure S6c displaces the cumulative heat flow applied to or extracted from the system. Linear fitting is applied to obtain the heat flow rate. The heat flow rate and temperature difference on both sides are averaged when evaluating the thermal conductivity.

Fig. S6 | NEMD setup and results. **a** Dimensions of the NEMD simulation system, and the hot and cold reservoirs in the x - y plane. **b** Temperature gradient along the x direction for the $\delta = 0.25$ case. **c** Cumulative thermal energy in the hot and cold reservoirs.”

In addition, we have updated Fig. 3d and related discussions in the revised manuscript to include the NEMD calculation results:

“To further understand the trend of Λ_{lat} versus V_g (and therefore also δ), we performed Green-Kubo MD (GKMD) and nonequilibrium MD (NEMD) simulations of thermal conductivity (see Supplementary Section 5 for calculation details), with results shown in Fig. 3d. The GKMD and NEMD results agree reasonably well with each other.

Fig. 3 | Impact of V_g on thermal properties. **A** The thermal conductivity (left axis) and electrical resistivity (right axis) of LSCO films, **b** the interfacial thermal conductance between Pt and LSCO, and **c** oxygen non-stoichiometry due to vacancies, as functions of V_g . In panel **a**, Λ_{tot} represents the measured total thermal conductivity of the LSCO films. Λ_{el} represents the electronic thermal conductivity estimated from electrical conductivity based on the Wiedemann-Franz law. The black dash-dotted lines represent the upper and lower limits of Λ_{el} based on different Lorenz numbers (L). The dashed lines serve as guides to the eye. In panel **c**, $\delta = 0.5$ (blue circles) is assumed for the complete transition to brownmillerite ($V_g \geq 3$ V), as our method used to estimate δ (red squares) is not valid above $\delta \approx 0.25$ (see Methods). The red-blue graded dashed line serves as a guide to the eye. **d** MD simulation results, obtained from both GKMD and NEMD, for the lattice thermal conductivity of LSCO with different δ . In all panels, the vertical pink and blue dashed lines represent, respectively, the starting and ending points of the P \rightarrow P + BM \rightarrow BM transformations. **e** Comparison of thermal conductivity tuning factor through bi-state tuning process in this work and previous experimental studies^{10,19-28}, including $\text{PbZr}_{0.3}\text{Ti}_{0.7}\text{O}_3$ (PZTO, 5 cycles)^{20,21}, PbZrO_3 (PZO, 10+ cycles)²², PbTiO_3 (PTO)²³, BiFeO_3 (BFO)²⁴, bio-polymers (B-P, 10+ cycles)²⁵,

azobenzene polymers (A-P, 6 cycles)²⁶, liquid crystal networks (LCN)²⁷, Li_xCoO₂ (LCO, 2 cycles)¹⁹, WO₃ (WO, 3 cycles)²⁸, and SrCoO_{3-δ} (SCO, 1 cycle, and H-SCO)¹⁰. Here, one cycle is defined as the transformation from state/phase A to state/phase B, and then return to state/phase A. Cycle numbers are provided when available from the literature.”

For the sensitivity of the interfacial thermal conductance to the measurement, it will be clearer if the sensitivity curve is presented. Are there any known methods to verify the correctness of such a multivariate fit?

Response: Generally, the sensitivity analysis of TDTR measurements is a very robust and routine way to help extract the parameters of interest. For a certain parameter, the higher the sensitivity value, the lower the measurement uncertainty. If the measurement sensitivity to a parameter is lower than 0.01, we would not trust the reliability of fitting results for that particular parameter due to the large uncertainties. The sensitivity plots of the interfacial thermal conductances (G_1 and G_2) have already been provided in Fig. S3 of the original Supplementary Information. However, we notice that we did not include G_1 and G_2 in the figure caption previously, which might have caused confusion for Reviewer 3. We have therefore updated the caption of Fig. S3 on Page 7 of the revised Supplementary Information as follows:

Fig. S3 | Representative sensitivity analyses. Sensitivity plots of TDTR measurements for the ungated P-LSCO sample calculated with the following parameters: heat capacities (C_{Pt} , C_{LSCO} , C_{LAO}), film thicknesses (t_{Pt} , t_{LSCO}), thermal conductivities (Λ_{Pt} , Λ_{LSCO} , Λ_{LAO}), interfacial thermal conductances (G_1 and G_2), and the beam spot size of $12 \mu\text{m}$. Panels **a-c** are routine TDTR ratio sensitivity to multiple parameters with different modulation frequencies. Panel **d** is the dual-frequency measurement sensitivity. Solid lines mean the sensitivity value is positive, while dashed lines represent negative sensitivity values.

As discussed in the Supplementary Information, we designed the experimental conditions to ensure sufficiently large sensitivity to the parameters of interest. Besides, we also compared our G_1 and G_2 results with literature values and obtained reasonably good agreements as shown in Fig. S4 of the Supplementary Information:

“To validate our measured values of G_1 for BM-LSCO films exhibiting insulating behavior, we list G values for interfaces between Pt and different insulating oxides¹¹⁻¹⁴ in Fig. S4, which vary from 180 to 330 $\text{MW m}^{-2} \text{K}^{-1}$. Our measurement results for G_1 between Pt and BM-LSCO samples (gated at 3, 3.5 and 4 V with negligible electronic contribution) are determined to be 160, 150 and 200 $\text{MW m}^{-2} \text{K}^{-1}$, respectively, in good agreement with literature data.

Fig. S4 | Interfacial thermal conductance. Summary of the interfacial thermal conductance between Pt and oxides plotted against the Pt thickness. The G values of Pt/SiO₂ and Pt/Al₂O₃ are taken from the literature¹¹⁻¹⁴.

Therefore, we believe the sensitivity analysis presented in this work is robust and thus the fitting results from our data reduction are reliable.

Overall, I think the results are sound and interesting for the scientific community.

Response: We thank Reviewer 3 for the supportive overall comment.

REVIEWERS' COMMENTS

Reviewer #3 (Remarks to the Author):

The authors have added data to address my questions. I recommend publication.

Responses to review comments on NCOMMS-22-42600B

We are resubmitting our revised manuscript entitled “Wide-range continuous tuning of the thermal conductivity of $\text{La}_{0.5}\text{Sr}_{0.5}\text{CoO}_{3-\delta}$ films *via* room-temperature ion-gel gating”, after making the final round of revisions based on the editorial suggestions/requests and comment from Reviewer 3.

Please refer to the cover letter for a detailed list of the materials required for the editorial review.

Reviewer #3 (Remarks to the Author):

The authors have added data to address my questions. I recommend publication.

Response: We are delighted that Reviewer 3 is satisfied with our revisions, and we appreciate the supportive comments.